

# Four-quark scatterings in QCD I

Wei-jie Fu[1], Chuang Huang[1*], Jan M. Pawlowski[2,3] and Yang-yang Tan[1]

1 School of Physics, Dalian University of Technology, Dalian, 116024, P.R. China
2 Institut für Theoretische Physik, Universität Heidelberg,
Philosophenweg 16, 69120 Heidelberg, Germany
3 ExtreMe Matter Institute EMMI, GSI, Planckstraße 1, D-64291 Darmstadt, Germany

★ huangchuang@mail.dlut.edu.cn

## Abstract

We investigate dynamical chiral symmetry breaking and the emergence of mesonic bound states from the infrared dynamics of four-quark scatterings. Both phenomena originate from the resonant scalar-pseudoscalar channel of the four-quark scatterings, and we compute the functional renormalisation group (fRG) flows of the Fierz-complete four-quark interaction of up and down quarks with its $t$ channel momentum dependence. This is done in the isospin symmetric case, also including the flow of the quark two-point function. This system can be understood as the fRG analogues of the complete Bethe-Salpeter equations and quark gap equation. The pole mass of the pion is determined from both direct calculations of the four-quark flows in the Minkowski regime of momenta and the analytic continuation based on results in the Euclidean regime, which are consistent with each other.



## 1 Introduction

The dynamical emergence of masses and spectra is one of the most intriguing questions in particle physics. Although the current quark masses are generated by the Higgs mechanism in the Standard Model of particle physics [1], they only account for $\sim 2\%$ of the proton mass. The missing $\sim 98\%$ mass of the proton, or more generally most of the mass of visible matter in the universe originates from dynamical strong chiral symmetry breaking. The respective pseudo-Goldstone bosons are the composite pions that acquire their masses from the current $u$, $d$ quark masses, and are massless in the chiral limit [2].

In the past two decades impressive progress has been made both within continuum QCD with functional approaches, see [3–7], and lattice simulations, [8–10]. In functional QCD one typically evaluates bound state equations such as Bethe-Salpeter equations [11,12] (two-body), Faddeev equations [13–15] (three-body) and four-body equations, utilising results for quark and gluon correlation functions obtained in 1st principle functional QCD, for recent reviews see e.g. [3,16]. Commonly, such a combination is used self-consistently in particular for the sake of symmetry identities and the persistence of massless states throughout the system, while also allowing for an access to timelike momenta. This combination of non-trivial requirements poses restrictions on such self-consistent approximations, which lead to further approximations both in the bound state sector and in the functional equations for quark and gluon correlation functions.

In the current work we complement the functional renormalisation group approach for bound state computations in a QCD set-up in [5, 17–22]. This approach is based on dynamical hadronisation, [5, 18, 22–26], which utilises the option to describe specific tensor and momentum channels in scattering vertices by the exchange of effective fields.

If restricting ourselves to momentum-independent tensors, we are left with a Fierz-

complete basis of ten tensor structures. The full basis including momentum-dependent tensor structures is much larger, see [3]. In most of the above works only the scalar-pseudoscalar channel ($\sigma$-pion) of the four-quark interaction is treated with dynamical hadronisation, as this channel carries the lightest degrees of freedom, the pions. We note, that within this setup functional QCD flows naturally into chiral perturbation theory at low momentum scales.

The exchange of the effective low energy fields take into account one momentum channel (in the above case the $t$-channel), which leaves us with a remnant four-quark interaction in this tensor channel, cf., e.g. [27, 28]. This remnant interaction in the scalar-pseudoscalar channel can be now iteratively treated with further momentum channel fields, as can be also introduced for other tensor structures or higher scattering vertices, see [22].

While possible, we consider it in most cases more efficient to keep both the remnant scalar-pseudoscalar vertex as well as all the other tensor channels as four-quark vertices and evaluate their full momentum dependence. Subject to a sufficiently quantitative approximation such an approach also allows to detect further emergent resonances as resonant channels in the four-quark vertices. Then, potentially also these resonances may or may not be treated with dynamical hadronisation, depending on their scattering dynamics. For spacelike (Euclidean) momenta it has been shown that the $t$-channel of the scalar-pseudoscalar channel of the four-quark vertex is by far dominating the flows as well as the physics [18,20], which is at the root of the success of chiral perturbation theory, and the rapid convergence of functional computations of Euclidean correlation functions in QCD. In turn, for timelike momenta or finite chemical potential we expect that both the general momentum dependence of the scalar-pseudoscalar channel as well as further channels become relevant or even dominating. Here, a relevant example is the diquark channel which is known to play an important rôle in Faddeev equations for baryons [3] as well as QCD at large density [29–31]. In summary, it is suggestive to keep as much tensor structures and momentum dependence as possible of the full coupled system of four-quark scattering vertices, thus monitoring the emergence of resonant tensor structures as well as non-trivial momentum dependences.

In the present work, we initiate the latter important endeavor by studying the flow of the coupled set of the vertex dressings of the ten Fierz complete four-quark interactions in two-flavour QCD in the chiral limit, accompanied by the flow of the (inverse) quark propagator.

In a first step we only consider the system of scale-dependent vertices and quark propagator, for some earlier related work on Fierz complete systems in QED and QCD at vanishing and finite temperature see [32–35, 35–38], for a review see [39]. This approximation allows us to study dynamical chiral symmetry breaking for generic quark masses, including the chiral limit within an extrapolation: We show that the system can be solved for all cutoff scales except for the chiral limit with vanishing current quark masses. In the chiral limit it develops an unphysical divergence of the constituent quark mass. This behaviour sets in for very small current quark masses (or rather pion masses), and the results admit a physical extrapolation to the chiral limit.

In a second step we also consider the $t$-channel momentum dependence of the scalar-pseudoscalar channel in a Fierz complete system. This allows us to solve the system for Euclidean spacelike momenta as well as timelike momenta; which gives us access to the pion pole mass as the resonance pole in the pseudoscalar channel of the four-quark scattering vertex. This result of the direct timelike (Minkowski space) computation agrees well with the extrapolation result also computed here.

A study with the full momentum dependences of the $s, t, u$ channels as well as the discussion of the remnant momentum dependence is subject of ongoing work, [40], as is the embedding in QCD, [41]. While we could treat the scalar-pseudoscalar channel with dynamical hadronisation, we refrain from doing so in order to evaluate the systematics of the four-quark flows as well as the emergence of the resonant interaction in the four-quark representation. In

consequence the current $t$-channel setup is tantamount to that used in [18,42], if dropping the multi-scattering of the pions and the $\sigma$-mode and considering only the cutoff scale dependence of the remaining channels (pointlike dispersion). This naturally allows for an embedding of the present flows and results and that in [40] within QCD [41].

This work is organised as follows: In Section 2 we discuss our approximation for the effective action with a quark two-point function and the Fierz-complete four-quark interaction, including the respective set of coupled flow equations. In Section 3, dynamical chiral symmetry breaking and the quark mass generation are investigated. In Section 4 we discuss the natural emergence of bound states. In Section 5 we summarise and discuss our findings. Some technical details are presented in appendices.

## 2 Functional RG approach to QCD and four quark scatterings

In the present work and the following ones, [40,41], we aim at a comprehensive description of the infrared scattering physics of quarks, as well as laying the foundations for further ones. This line of research is embedded in the general research line of the fQCD collaboration, aiming at the description of QCD at finite temperature and density with functional methods, including hadron resonances and their formation in the medium. Therefore we use this Section also to briefly review the general setup and some convention uses in our collaboration.

We use the functional renormalisation group approach to QCD, in which the evolution of the full effective action of QCD is described by its functional flow equation for the effective action $\Gamma_k[\Phi]$ where $k$ is an infrared cutoff scale that regularises the infrared propagation of all fields, for recent reviews see [43,44]. Including dynamical hadronisation, the flow equation reads [5,45],

$$\partial_t \Gamma_k[\Phi] + \int \dot{\phi}_{k,i}[\Phi] \left( \frac{\delta \Gamma_k[\Phi]}{\delta \phi_i} + c_\sigma \delta_{i\sigma} \right) = \frac{1}{2} \text{Tr}\left( G_k[\Phi] \partial_t R_k \right) + \text{Tr}\left( G_{\phi \Phi_j}[\Phi] \frac{\delta \dot{\phi}_{k,i}[\Phi]}{\delta \Phi_j} R_\phi \right), \quad \text{(1a)}$$

with the RG time $t = \ln(k/\Lambda)$, and

$$G_k[\Phi] = \frac{1}{\Gamma_k^{(2)}[\Phi] + R_k}, \qquad G_{\Phi_i \Phi_j} = (G_k)_{\Phi_i \Phi_j}. \quad \text{(1b)}$$

In (1a) we have introduced the superfield $\Phi$ with

$$\Phi = (\Phi_f, \phi), \qquad \Phi_f = (A, c, \bar{c}, q, \bar{q}), \qquad \phi = (\sigma, \pi), \quad \text{(1c)}$$

where $\Phi_f$ contains the fundamental fields in QCD, $\phi$ comprises the effective degrees of freedom, and $\dot{\phi}_k[\Phi]$ denotes their change with $k$. As an example we have used the effective fields for the scalar-pseudoscalar channel ($t$-channel), the $\sigma$-mode and the pions $\pi$ in (1c). These are the commonly used effective degrees of freedom, but the setup is not restricted to this, see [22]. General (1PI) correlation functions are denoted by

$$\Gamma_{\Phi_{i_1} \cdots \Phi_{i_n}}^{(n)}(\boldsymbol{p}) = \frac{\delta}{\delta \Phi_{i_n}} \cdots \frac{\delta \Gamma_k[\Phi]}{\delta \Phi_{i_1}}, \qquad \boldsymbol{p} = (p_1, ..., p_n), \quad \text{(1d)}$$

with $\Phi_{i_j} = \Phi_{i_j}(p_j)$ and all momenta are taken as incoming, which are depicted in Figure 1. Also, the vertex $\Gamma^{(n)}$ is taken in a general background $\Phi$.

In the present work we consider two-flavour QCD, and hence $q = (u, d)$. In (1) we have introduced the effective fields for the scalar-pseudoscalar channel ($t$-channel), the $\sigma$-mode

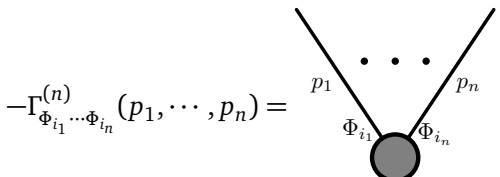

Figure 1: Diagrammatic representation of general 1PI $n$-point functions or vertices, as shown in (1d).

and the pions $\pi$. More details can be found in [5], or the reviews [43, 44]. This QCD set-up will be used in the subsequent works [40] and in particular [41], while in the present work we concentrate on the matter sector in the infrared regime for cutoff scales $k \lesssim 1$ GeV. Accordingly we drop the gluons, which indeed decouple successively in this regime. We note that one may also integrate out the gluons which leaves us with a non-local effective action of the remaining matter degrees of freedom. Accordingly, this setup still allows for qualitative and quantitative computations depending on the choice of the initial effective action at $k \approx 1$ GeV, for a detailed discussion see [5]. Finally, we refrain from using dynamical hadronisation in the scalar-pseudoscalar channel as we want to investigate the persistence and reliability of computations in fundamental quark correlation functions.

## 2.1 Matter sector of QCD and four quark scatterings

The restriction to the infrared matter sector of QCD described above leaves us with an effective action of the quarks, which can be expanded in powers of the quark and anti-quark field $q, \bar{q}$. In the vacuum the higher order scatterings $(\bar{q}q)^n$ for $n \geq 3$ are strongly suppressed and we do not consider them here, which leaves us with the kinetic term of the quarks $\Gamma_{\text{kin}}$ and the four-quark scattering term, $\Gamma_{4q}$ with

$$\Gamma_k[q, \bar{q}] = \Gamma_{\text{kin},k}[q, \bar{q}] + \Gamma_{4q,k}[q, \bar{q}]. \tag{2}$$

The general kinetic term reads

$$\Gamma_{\text{kin},k} = \int_{x,y} Z_q(x, y) \bar{q}(x) \left[ \slashed{\partial} + M_q(x, y) \right] q(y), \tag{3}$$

with $\int_x = \int d^4 x$, and

$$\slashed{\partial} = \gamma_\mu \partial_\mu, \quad \text{and} \quad \{\gamma_\mu, \gamma_\nu\} = 2\delta_{\mu\nu}. \tag{4}$$

Both, $Z_q$ and $M_q$ carry also a $k$-dependence. The general kinetic term (3) also accommodates breaking of translation invariance as may happen in inhomogeneous phases or moat regimes [46], or in expanding backgrounds.

In the present work we perform computations in the vacuum, and hence (3) reduces to

$$\Gamma_{\text{kin},k} = \int_p Z_q(p) \bar{q}(-p) \left[ i\slashed{p} + M_q(p) \right] q(p), \tag{5}$$

with $\int_p \equiv \int d^4 p/(2\pi)^4$. The general four quark scattering term is given by

$$\Gamma_{4q,k} = \int_{\boldsymbol{p}} \lambda_\alpha(\boldsymbol{p}) \mathcal{T}^{(\alpha)}_{ijlm}(\boldsymbol{p}) \bar{q}_i(p_1) \bar{q}_j(p_2) q_l(p_3) q_m(p_4), \tag{6}$$

$$\Gamma^{(4)}_{\bar{q}_i\bar{q}_j q_l q_m}(p_1,p_2,p_3,p_4) = \quad$$ 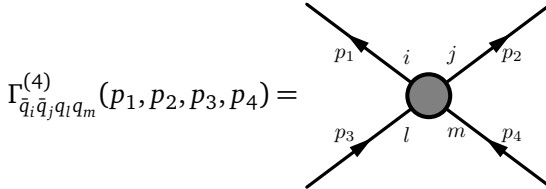

Figure 2: Diagrammatic representation of the quark four-point function. All momenta are counted as incoming and $i,j,l,m$ carry the Dirac, flavour and color indices, see the representation of general 1PI $n$-point functions in Figure 1.

where a sum over $\alpha = 1,...,512$ is implied, and

$$\int_{\boldsymbol{p}} \equiv \int \frac{d^4 p_1}{(2\pi)^4}\cdots\frac{d^4 p_4}{(2\pi)^4}(2\pi)^4\delta^4(p_1+\cdots+p_4). \tag{7}$$

The set of tensors $\{\mathcal{T}^{(\alpha)}_{ijlm}(\boldsymbol{p})\}$ with $\alpha = 1,...,512$ constitutes a complete set of four-quark tensor structures, see e.g. [47]. The indices $i,j,l,m$ are composed from Dirac, flavour ($N_f = 2$) and color ($N_c = 3$) indices. We have suppressed the field indices as well as the cutoff dependence in the four-quark couplings, $\lambda_\alpha = \lambda^{(\alpha)}_{qq\bar{q}\bar{q},k}$, as the superscript $(\alpha)$ already labels the basis. We can project onto the different tensor structures or rather their $k$-dependent dressings $\lambda_\alpha(\boldsymbol{p})$ by considering

$$\Gamma^{(4)}_{\bar{q}_i\bar{q}_j q_l q_m}(\boldsymbol{p}) = 4\lambda_\alpha(\boldsymbol{p})\,\mathcal{T}^{(\alpha)}_{ijlm}(\boldsymbol{p})(2\pi)^4\delta^4(p_1+\cdots+p_4), \tag{8}$$

and contracting (8) with the $\mathcal{T}^{(\alpha)}_{ijlm}$, where the factor 4 on the r.h.s. arises from the two quark fields and two anti-quark fields. In (8), all momenta are counted as incoming, thus leading to the sum of all momenta in the $\delta$-function that carries momentum conservation. The symmetries of the couplings $\lambda_\alpha(\boldsymbol{p})$ under commutation of the momentum arguments follow from that of the respective tensors $\mathcal{T}^{(\alpha)}_{ijlm}$ and crossing symmetry. Equation (8) is depicted in Figure 2.

This general setup is important for resonance computations, while most of the four-quark tensor structures in (6) are negligible for the off-shell dynamics of the theory. In particular they effectively decouple from the dynamics of the lower order correlation functions that drive the chiral and confinement dynamics of QCD. The fRG approach used here is tailor-made for accommodating these properties explicitly: Firstly the momentum loops in (1) only carry (off-shell) loop momenta with $p^2 \lesssim k^2$. This allows for an efficient, rapidly converging expansion of loops in external momenta. In particular this suppresses the contributions of vertices that vanish at $\boldsymbol{p} = 0$. Secondly, the infrared suppression of momentum modes with $p^2 \lesssim k^2$ in the propagators suppresses the effects of angular dependences by the respective angular averages, hence further suppressing the contributions of momentum-dependent tensor structures. In summary, we expect a rapidly converging expansion of (6), if the tensor structures are ordered in terms of powers of momenta. Note that this implies an expansion in *smooth* tensor structures without kinematic divergences, for related and more general discussions see e.g. [16, 20, 48].

This finalises our discussion of the four-quark scattering part in the effective action in $N_f = 2$ flavour QCD.

## 2.2 Flow of quark propagator and four-quark vertices

We proceed with the system of flow equations of the four-quark vertices. In QCD the four-quark vertices are generated by quark-gluon diagrams that dominate the respective flows for

$$-\Gamma_{\bar{q}q}^{(2)}(p',p) =$$

Figure 3: Diagrammatic representation of the quark two-point function. All momenta are counted as incoming, see the representation of general 1PI $n$-point functions in Figure 1.

cutoff scales $k \gtrsim 1\,\text{GeV}$. For smaller scales, $\lesssim 1\,\text{GeV}$ the dynamics of the pure matter sector is taking over, and there the flow is dominated by that governed by (2) with (5) and (6). This leads us to the coupled flows for the quark propagator and the four-quark vertices depicted in Figure 4. The first flow is that of the quark two-point functions in a vanishing background $\Phi = 0$,

$$\Gamma_{\bar{q}q,k}^{(2)}(p',p) = Z_q(p)\Big[\,\mathrm{i}\slashed{p} + M_q(p)\Big](2\pi)^4\delta(p+p'),\tag{9}$$

where both momenta are incoming as defined in (1d) for general $n$-point correlation functions. The two-point function (9) is depicted in Figure 3.

The full quark two-point function is given by the sum of $\Gamma_{\bar{q}q}^{(2)}$ and the regulator $\mathcal{R}_q$

$$\mathcal{R}_q = \begin{pmatrix} 0 & R_{q\bar{q}} \\ R_{\bar{q}q} & 0 \end{pmatrix}, \qquad R_{\bar{q}q} = -R_{q\bar{q}}^T,\tag{10a}$$

with the superscript $T$ denoting the transpose and the chiral regulator that preserves chiral symmetry, reading

$$R_{\bar{q}q} = Z_q(p)\,\mathrm{i}\slashed{p}\,r_q(x), \qquad x = \frac{p^2}{k^2}, \qquad \text{or}\quad p \to \boldsymbol{p}\,.\tag{10b}$$

The substitution of the four momentum with its spatial part also implies $\slashed{p} \to \boldsymbol{p}\,\boldsymbol{\gamma}$. In short, we allow for chiral spatial momentum ($3d$) and full momentum ($4d$) regulators. These options as well as a variation of the shape is used for a reliability check of the results obtained here. For this check see Appendix G.

With the wave function factor in (10b) the regulator is RG-adapted [25]: These regulators preserve the underlying RG-scaling of the physical theory without the regulator. Moreover, this choice leads to a uniform suppression of momentum modes in all the tensor structures of the regulator, see (12).

In (10b) we have introduced the dimensionless shape function $r_q(x)$ with $x = p^2/k^2$. In the present work we consider general shape functions, but a common simple choice is given by the shape function of the flat or Litim regulator, [49,50], see Appendix G. In summary, the quark propagator $G_{q\bar{q}}(p,p')$ in homogeneous backgrounds is given by

$$G_{q\bar{q}} = \left[\frac{1}{\Gamma_k^{(2)} + \mathcal{R}_q}\right]_{q\bar{q}} = G_q(p)(2\pi)^4\delta^4(p+p'),\tag{11}$$

where we have suppressed the momentum arguments in the first two expressions. The

$$\partial_t \left( \rightarrow\!\!\bullet\!\!\rightarrow \right) = \tilde{\partial}_t \left( -\, \bigcirc \right)$$

$$\partial_t \left( \times\!\!\bullet\!\!\times \right) = \tilde{\partial}_t \left( -\, \times\!\!\bullet\!\!\times + \times\!\!\bullet\!\!\times + \frac{1}{2}\, \times\!\!\bigcirc\!\!\times \right)$$

Figure 4: Diagrammatic representation of the flow equations for the two-point and four-point correlation functions. Here $t = \ln(k/\Lambda)$ is the RG time with a UV cutoff $\Lambda$. The partial derivative with a tilde denotes that it only hits the dependence of the RG scale through the regulator in propagators, whose implementation would result in the insertion of a regulator for every inner line of diagrams on the r.h.s. of flow equations.

momentum dependent kernel is given by

$$G_q(p) = \frac{1}{Z_q(p)} \frac{1}{i\slashed{p}\left[1 + r_q\left(p^2/k^2\right)\right] + M_q(p)} \,. \tag{12}$$

With the RG-adapted choice of the regulator in (10b) the propagator has a uniform dependence on the quark wave function $Z_q(p)$ for all cutoff scales. The shape function $r_q$ modifies the classical Dirac dispersion $\slashed{p}$ not involving the wave function. This is at the core of the RG-adaptation introduced in [25].

These preparations allow us to consider the full coupled set of flow equations of the general effective action (2) of the quark sector. The flow equations for the inverse quark propagator and the four-quark vertices are depicted in Figure 4, see Equations (B.1) in Appendix B. In the following sections, we will employ these flow equations within different approximations to the full set of four-quark vertices for the investigation of dynamical chiral symmetry breaking, Section 3, and the emergence of bound states, Section 4.

## 2.3 Flow of off-shell relevant correlation functions

The successive momentum shell integration of the flow with Euclidean (spacelike) loop momenta $q^2 \lesssim k^2$ allows for a very effective relevance ordering within the coupled set of flow equations for vertex and propagator dressing. To begin with, all four quark channels decouple very rapidly for $k \approx 1\,\text{GeV}$ towards larger cutoff scales. This reflects the dominance of (off-shell) quark-gluon fluctuations in this regime. In turn, the chiral symmetry breaking cutoff scale is $k_\chi \approx 0.5\,\text{GeV}$, below which we expect resonant interaction channels.

The investigation of dynamical chiral symmetry breaking only requires a quantitative grip on the off-shell quark two-point function. In view of the relevance ordering of the complete set of tensor structures $\{\mathcal{T}_{ijlm}^{(\alpha)}(\boldsymbol{p})\}$ of the four-quark vertex discussed at the end of Section 2.1 below (8), we only consider the set of momentum-independent tensor structures, $\{\mathcal{T}_{ijlm}^{(\alpha)}\}$, see Appendix A.

For two flavours such a Fierz-complete basis has ten basis elements, and a specific one is listed in Appendix A. For 2+1 flavours the basis has 26 basis elements, see e.g. [51]. Note that the tensor structures $\mathcal{T}_{ijlm}^{(\alpha)}$ carry the symmetries of the combination of the four Grassmann variables (momentum-independent), with which they are contracted. Comprehensive discussions concerning projections, flows and partial momentum dependences can be found e.g. in [18, 20, 31, 39, 51].

These considerations lead us to a reduced but still rather general four quark sector, whose effective action includes the kinetic term (5) and four-quark terms with a Nambu–Jona-Lasinio type structure [2, 52],

$$\Gamma_{4q,k} = \int_{\boldsymbol{p}} \lambda_\alpha(\boldsymbol{p}) \, \mathcal{T}^{(\alpha)}_{ijlm} \, \bar{q}_i(p_1) \bar{q}_j(p_2) q_l(p_3) q_m(p_4) \, , \tag{13}$$

with $\alpha = 1, ..., 10$, indicating the tensor basis discussed in Appendix A. Alternatively, using labels for the basis elements there, one arrives at

$$\alpha \in \left\{ (V \pm A), \, (S \pm P)_\pm, \, (V - A)^{\text{adj}}, \, (S \pm P)^{\text{adj}}_-, \, (S + P)^{\text{adj}}_+ \right\}. \tag{14}$$

The Fierz complete tensors $\mathcal{T}^{(\alpha)}_{ijlm}$ are antisymmetric under the commutation of pairs of indices $i, j$ and $l, m$, see (A.2) in Appendix A. Accordingly, the ten vertex dressings $\lambda^{(\alpha)}(\boldsymbol{p})$ are symmetric under the interchange of the momenta $p_1$ and $p_2$, or that of $p_3$ and $p_4$, i.e.,

$$\lambda_\alpha(p_1, p_2, p_3, p_4) = \lambda_\alpha(p_2, p_1, p_3, p_4) = \lambda_\alpha(p_1, p_2, p_4, p_3) = \lambda_\alpha(p_2, p_1, p_4, p_3). \tag{15}$$

In Appendix F we discuss a possibility to extend the ten four-quark dressings to another ten ones, which are anti-symmetric under the interchange of two quarks or antiquarks.

In our computation we re-arranged the tensor elements, $\mathcal{T}^{(S-P)_+}$ in (A.1a), $\mathcal{T}^{(S+P)_-}$ in (A.1b), $\mathcal{T}^{(S-P)_-}$ in (A.1c), and $\mathcal{T}^{(S+P)_+}$ in (A.1d), such that they carry either scalar or pseudoscalar quantum numbers. This facilitates the identification of scalar or pseudoscalar resonances from momentum channels of a given tensor structure. We are led to

$$\mathcal{T}^\sigma_{ijlm} \bar{q}_i q_l \bar{q}_j q_m = (\bar{q} \, T^0 q)^2 \, ,$$

$$\mathcal{T}^\pi_{ijlm} \bar{q}_i q_l \bar{q}_j q_m = -(\bar{q} \, \gamma_5 T^a q)^2 \, ,$$

$$\mathcal{T}^a_{ijlm} \bar{q}_i q_l \bar{q}_j q_m = (\bar{q} \, T^a q)^2 \, ,$$

$$\mathcal{T}^\eta_{ijlm} \bar{q}_i q_l \bar{q}_j q_m = -(\bar{q} \, \gamma_5 T^0 q)^2 \, . \tag{16}$$

It is left to extract the coupled flows of the ten respective vertex dressings as well as those of the quark wave function and the quark mass function from the flows in Figure 3 and Figure 4. Contracting the flow of the four-point correlation functions with all ten tensor elements provides us with the flow of all couplings. Contracting the flow of the quark two-point function with $\not{p}$ and $\mathbb{1}$ leads us to the flow of the quark wave function, the quark mass function respectively. The required projections and contractions of the flows are done with the aid of FormTracer, a Mathematica tracing package using FORM, see [53] for details.

The current work initiates the comprehensive investigation of the flow equations (B.1) and their solution with different approximations, e.g., with or without the momentum dependence, to investigate the mechanism of quark mass production and the natural emergence of bound states, and also to extend the relevant flow equations to include the glue dynamics such that the low energy effective theory is extended to a first-principle QCD, see e.g., [5, 17–20].

## 3 Chiral symmetry breaking and quark mass production

In this section we use the present four-quark fRG set-up to access the mechanism of the emergence of the constituent quark mass due to the dynamical chiral symmetry breaking. In a first

step we neglect the momentum dependence of the four-quark couplings and the quark mass,

$$\lambda_\alpha = \lambda_\alpha(p_i = 0), \qquad (i = 1, \cdots, 4),$$

$$M_q = M_q(p = 0), \tag{17}$$

and assume the quark wave function $Z_q = 1$, which simplifies the numerical calculations significantly. Obviously, this truncation is only a very qualitative one, and we expect it to break down in the chiral limit: there, $\lambda_\alpha(p)$ is bound to develop a pole and it is suggestive that in a momentum-independent approximation this pole will be a global non-integrable singularity for all momenta. Indeed, this is precisely what happens, see Figure 6 and the respective discussion. Still, the approximation should work well away from this singularity, and one of the aims of the present work and the following ones is to find out minimal truncations that capture qualitative and quantitative physics.

For the following computations we introduce dimensionless variables for convenience, and in particular we shall use

$$\bar{\lambda}_\alpha = \lambda_\alpha k^2, \qquad \bar{M}_q = \frac{M_q}{k}. \tag{18}$$

## 3.1 Single channel approximation

A further simplification is achieved by assuming the dominance of the scalar-pseudoscalar channel and hence dropping the other nine channels. This approximation is tailor-made for unravelling the dynamics of chiral symmetry breaking, also at work in the full system. Moreover, this has been proven as a very good approximation within full QCD flows, where the dominant scalar-pseudoscalar channel has been also treated with dynamical hadronisation, [20]. There it has been checked that the scalar-pseudoscalar channel is by far the dominant one by switching off one by one the other channels. Here we confirm this property within the four-quark setup. Moreover, we identify $\sigma$ and $\pi$ exchange. This is summarised in

$$\lambda_{\sigma-\pi} \equiv \lambda_\pi = \lambda_\sigma, \qquad \lambda_{\alpha \notin \{\sigma,\pi\}} = 0. \tag{19}$$

In this approximation, the full flow equations in (B.1c) and (B.1b) reduce to simple flows for the dimensionless coupling and quark mass,

$$\partial_t \bar{\lambda}_{\sigma-\pi} = 2\bar{\lambda}_{\sigma-\pi} + \frac{\bar{\lambda}_{\sigma-\pi}^2}{2\pi^2} \int_0^\infty dx\, x^3 r_q{}'(x) \times \frac{\left(-4\bar{M}_q^2 + 7x\left[1 + r_q(x)\right]^2\right)\left[1 + r_q(x)\right]}{\left(x\left[1 + r_q(x)\right]^2 + \bar{M}_q^2\right)^3}, \tag{20a}$$

and

$$\partial_t \bar{M}_q = -\bar{M}_q + \bar{M}_q \bar{\lambda}_{\sigma-\pi} \frac{13}{4\pi^2} \int_0^\infty dx\, x^3 r_q{}'(x) \times \frac{1 + r_q(x)}{\left(\left[1 + r_q(x)\right]^2 x + \bar{M}_q^2\right)^2}. \tag{20b}$$

The set of flow equations (20) are the standard flow equations in the local NJL-type model, for a detailed discussion of the flows in the chiral limit, and a first qualitative discussion of the mass-dependence see [39]. The flow for the coupling can be written schematically as

$$\beta_{\bar{\lambda}_{\sigma-\pi}} \equiv \partial_t \bar{\lambda}_{\sigma-\pi} = 2\bar{\lambda}_{\sigma-\pi} - \mathcal{C}(\bar{M}_q)\bar{\lambda}_{\sigma-\pi}^2, \tag{21}$$

with

$$\mathcal{C}(\bar{M}_q) = \frac{7 - 4\bar{M}_q^2}{8\pi^2\left(1 + \bar{M}_q^2\right)^3}, \tag{22}$$

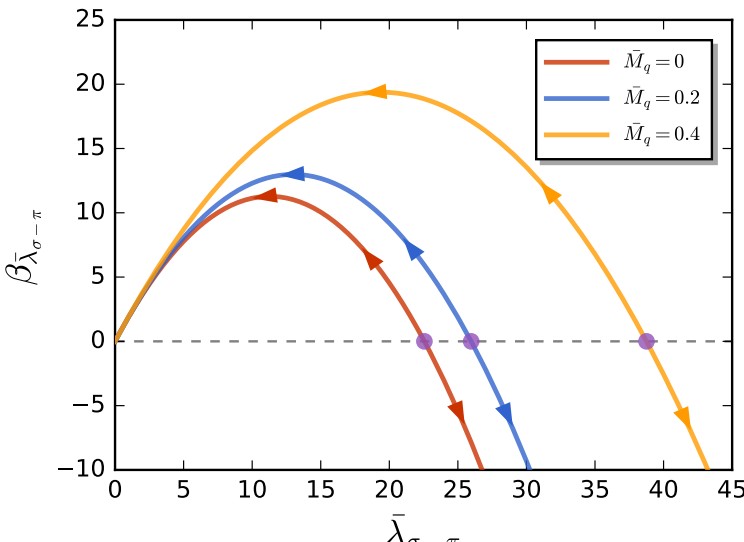

Figure 5: $\beta$-function $\beta_{\bar{\lambda}_{\sigma-\pi}}$ defined in (21) as a function of the coupling $\bar{\lambda}_{\sigma-\pi}$ with vanishing and finite dimensionless quark mass $\bar{M}_q$.

for the flat or Litim regulator (G.1). Fixed points of the flow are defined by $\beta_{\lambda_{\sigma-\pi}} = 0$. Apart from the Gaußian fixed point at $\lambda_{\sigma-\pi} = 0$, the system has a non-trivial fixed point,

$$\bar{\lambda}^*_{\sigma-\pi}(\bar{M}_q) = \frac{2}{\mathcal{C}(\bar{M}_q)} \, . \tag{23}$$

This fixed point is an attractive ultraviolet fixed point: wherever we initiate the flow towards larger cutoff scales, the coupling is drawn to this fixed point. For illustration we show in Figure 5 the $\beta$-function (21) for $\bar{M}_q = 0$, and these properties are clearly seen.

The location of this fixed point is regulator dependent. In Figure 5, where the single channel approximation with $\bar{M}_q = 0$ and the 4d flat regulator has been used, we obtain $\bar{\lambda}^*_{\sigma-\pi} = (16\pi^2)/7 \approx 22.56$ from (22). The fixed point location for all regulators used in the present work are summarised in Table 1 for both, the single channel and Fierz complete approximation.

The strong regulator dependence of the fixed point is not an artifact of the approximation and has two different but related sources [25]: First of all, the dimensionless coupling is rescales with $k^2$ and the relation of $k$ to a mass scale in the theory depends on the chosen regulator. Second, the regulator choice also implies a coice of RG scheme which shows in

Table 1: Ultraviolet fixed point location $\lambda^*_{\sigma-\pi}$ for different regulators in the single channel and Fierz-complete approximations.

| Regulator | single channel | $\bar{\lambda}^*$ | Fierz-complete |
|---|---|---|---|
| 3d flat | 16.91 | | 16.81 |
| 4d flat | 22.56 | | 22.43 |
| 3d exp | 6.15 | | 6.10 |
| 4d exp | 9.98 | | 9.92 |

the the values of non-universal couplings. In short, the fixed point location alone carries no physics and strongly depends on the regulator.

The qualitative picture is best evaluated in the chiral limit for $\bar{M}_q = 0$ as depicted in Figure 5, where we show the $\beta$-function (21) as a function of $\bar{\lambda}_{\sigma-\pi}$ in the chiral limit. Then, the flow of $\bar{M}_q$ vanishes identically above the scale of dynamical chiral symmetry breaking as it is proportional to $\bar{M}_q$, see (20b). Consequently we only have to discuss the $\beta$-function of the coupling. If initiating the flow at $k = \Lambda$ with $\bar{\lambda}_{\sigma-\pi} < \bar{\lambda}^*_{\sigma-\pi}$, the dimensionless coupling will flow into the Gaußian fixed point at $\bar{\lambda}^*_{\sigma-\pi} = 0$ without dynamical chiral symmetry breaking. In turn, if the initial coupling is larger than the UV fixed point coupling, $\bar{\lambda}_{\sigma-\pi} > \bar{\lambda}^*_{\sigma-\pi}$, the coupling grows in the IR flow and finally hits a singularity at $k_\chi$, which signals chiral symmetry breaking. This simple analysis already entails that we cannot flow to $k = 0$ in the chiral limit with $\bar{\lambda}_{\sigma-\pi} > \bar{\lambda}^*_{\sigma-\pi}$, and the flow terminates at the chiral symmetry breaking cutoff scale $k_\chi$.

However, the chiral limit is a special case, and for non-vanishing mass the sign of the coefficient $\mathcal{C}(\bar{M}_q)$ of the $\beta$-function depends on the size of $\bar{M}_q$. In particular, it is negative for sufficiently large masses $\bar{M}_q > \bar{M}_q^{\text{Gauß}}$. For the flat regulator (G.1) we find

$$\bar{M}_q^{\text{Gauß}} = \frac{\sqrt{7}}{2}. \tag{24}$$

Above this mass the Gaußian regime extends to infinity. While the value of $\bar{M}_q^{\text{Gauß}}$ depends on the chosen regulator as does the location of the fixed point, the existence of this regime is a physics feature of the system and holds true for all regulators. In conclusion, for masses $M_q \propto k^\gamma$, that do not decay with $\gamma > 1$ for $k \to 0$ (massless limit), the dimensionless mass $\bar{M}_q$ grows large and we enter the Gaußian regime. For $\bar{M}_{q,\Lambda} \neq 0$ at the initial scale, the mass is flowing with (20b), and schematically it reads

$$\partial_t \bar{M}_q = -\bar{M}_q \Big[ 1 + \bar{\lambda}_{\sigma-\pi} C(\bar{M}_q) \Big], \tag{25}$$

with

$$C(\bar{M}_q) = \frac{13}{16\pi^2 \left(1 + \bar{M}_q^2\right)^2}, \tag{26}$$

for the flat or Litim regulator (G.1). We remark that in more advanced schemes the dimension counting term in (21) acquires an anomalous part

$$2\bar{\lambda}_{\sigma-\pi} \to [2 + \mathcal{B}(M_q)]\bar{\lambda}_{\sigma-\pi}, \tag{27}$$

where $\mathcal{B}(M_q)$ accounts for the anomalous scaling of the quark fields. While this is important for a fully quantitative analysis, it does not change the qualitative structure of the flow.

Hence, the dimensionless mass is always increasing towards the infrared and the coupled system of flow equations for $\bar{\lambda}_{\sigma-\pi}, \bar{M}_q$ always enters the Gaußian regime below some cutoff scale $k_{\text{Gauß}}$. Accordingly, the flow of the dimensionful coupling and quark mass parameters freezes in below this scale.

For physical quark masses close to the chiral limit, the four-quark coupling grows very large before entering the Gaußian regime: in this regime with an explicit small chiral symmetry breaking the pion dynamics grows strong, and the scalar-pseudoscalar channel of the four-quark interaction with the coupling $\lambda_{\sigma-\pi}$ develops a pseudoscalar resonance which carries the properties of a pion exchange with a pion propagator

$$\frac{1}{P^2 + m_\pi^2}, \tag{28}$$

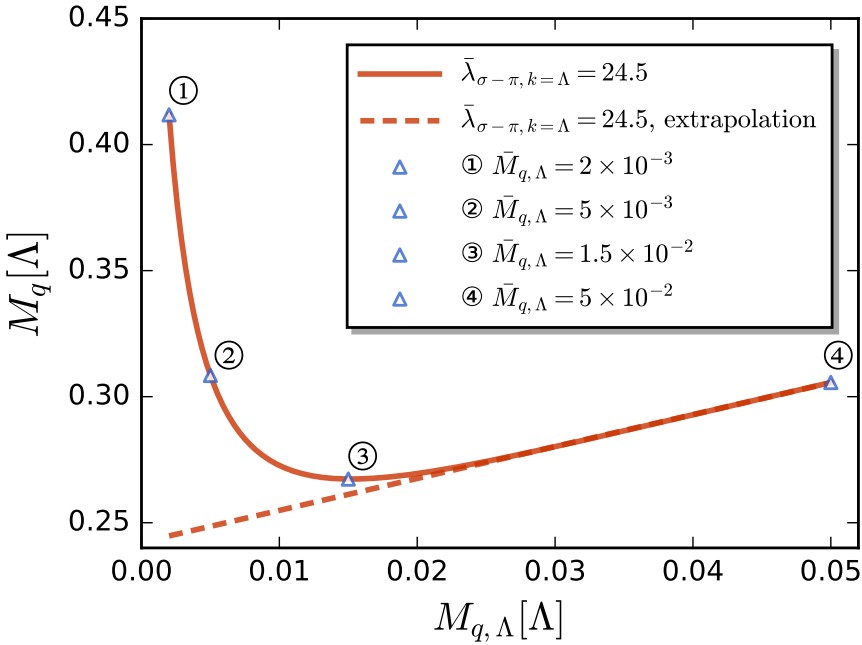

Figure 6: Physical constituent quark mass $M_{q,k\to 0}$ as a function of the UV (current) quark mass $M_{q,\Lambda}$ for an initial coupling $\lambda_{\sigma-\pi,\Lambda} = 24.5$. The results for $\bar{M}_{q,\Lambda}$ also depicted in Figure 7 are indicated on the curve in blue triangles. The dashed line denotes a linear extrapolation from the region of large current quark mass towards the chiral limit, and is supported by its presence within dynamical hadronisation, e.g. [54]. A comparison with the Fierz complete computation is done in Figure 12.

with the pion exchange momentum $P^2$. In the present momentum-independent approximation for $\lambda_{\sigma-\pi}$ the pion propagator in (28) is simply approximated by $1/m_\pi^2$ which greatly overestimates the strength of this channel for $m_\pi^2 \to 0$. In terms of the initial mass this regime close to the chiral limit is entered for

$$\bar{M}_{q,\Lambda} \lesssim \bar{M}_\chi \,, \tag{29}$$

and the latter has to be determined dynamically. This resonant behaviour of the pseudoscalar channel is also responsible for the singularity in the flow for $\bar{M}_q \equiv 0$, and it shows an unphysical growth of the constituent quark mass in the regime (29) if lowering the initial current quark mass $M_{q,\Lambda}$.

In this regime one either includes momentum-dependent four quark couplings and in particular $\lambda_{\sigma-\pi}(P)$ or one resorts to dynamical hadronisation as done in [5, 17, 18, 20, 54]. In the latter computations with dynamical hadronisation no unphysical rise of the constituent quark mass occurs in the chiral limit. and presence within dynamical hadronisation, e.g. [54]: the linear dependence on the current quark mass holds up into the chiral limit both in low energy effective theories and in QCD. In the present work we consider a momentum-dependent $\lambda_{\sigma-\pi}(P)$ for the discussion of bound states in Section 4.

The above structure, both the reliability of the present simple approximation for current quark masses $\bar{M}_{q,\Lambda} \gtrsim \bar{M}_\chi$, as well as the successive failure for $\bar{M}_{q,\Lambda} \lesssim \bar{M}_\chi$ is illustrated in Figure 6. There we show the physical constituent quark mass $M_{q,k\to 0}$ as a function of the initial (current) quark mass $M_{q,k=\Lambda}$ for an initial coupling value of $\bar{\lambda}_{\sigma-\pi} = 24.5$. The setups with the different initial masses $\bar{M}_{q,\Lambda}$ are indicated with numbers (in circles), counting from small initial masses close to the chiral limit to larger ones with a large explicit symmetry breaking.

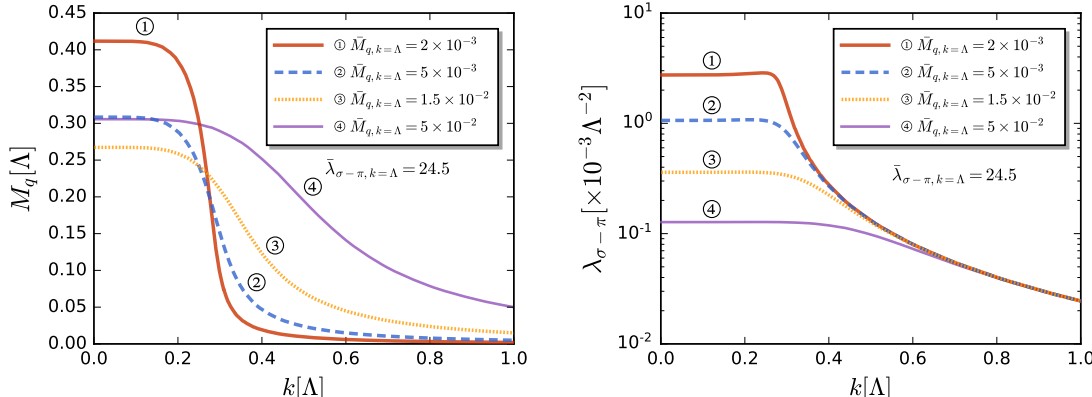

Figure 7: Quark mass $M_q$ (left panel) and four-quark coupling $\lambda_{\sigma-\pi}$ (right panel) as functions of the RG scale $k$ with different initial current quark mass values $\bar{M}_{q,\Lambda}$ and a fixed initial coupling $\bar{\lambda}_{\sigma-\pi,\Lambda} = 24.5$. The respective flowing masses $M_{q,k}$ and couplings $\lambda_{\sigma-\pi,k}$ are also indicated in Figure 6 and Figure 8.

The respective flowing masses $M_{q,k}$ and couplings $\lambda_{\sigma-\pi,k}$ are shown as functions of $k$ in Figure 7. There one clearly sees that the masses are monotonously decreasing with smaller $\bar{M}_{q,\Lambda}$ for $k \gtrsim 0.3\Lambda$. Already for $k \lesssim 0.3\Lambda$ the mass and coupling flows are too large due to the lack of momentum-dependence, and it is these oversized flows that cause the non-monotonicity of the constituent quark mass for small input masses. Note that this amplification of the flows for small cutoff scales is also present for larger initial quark masses, but there the flows are approximately vanishing for $k \lesssim 0.25\Lambda$, and hence this is irrelevant.

In our example the onset of the small mass regime, where the present simple approximation lacks reliability, is readily read off from Figure 6 as the initial mass $\bar{M}_{q,\Lambda}$, for which the curve stops being roughly linear in the initial mass. This leads us to

$$\bar{M}_\chi \approx 0.025, \qquad \text{for} \qquad \bar{\lambda}_{\sigma-\pi,\Lambda} = 24.5. \tag{30}$$

In summary this structural analysis leads to the following picture with three different regimes, the size of which is ruled by a characteristic mass scale $\bar{M}_\chi$ that depends on the initial coupling $\bar{\lambda}_{\sigma-\pi}$:

(i) $\bar{M}_{q,\Lambda} \gtrsim \bar{M}_\chi$: The system enters the Gaußian regime for sufficiently large $k$. Moreover, the quark mass and coupling also settle for large enough $k$ and the present approximation is working well.

(ii) $\bar{M}_{q,\Lambda} \lesssim \bar{M}_\chi$: The system enters the Gaußian regime for too small $k$, and the over-estimation of the infrared flows in the absence of momentum dependences has an increasing impact on the values of the physical masses and couplings at $k = 0$. In this regime a better momentum resolution is required to obtain accurate results, as is discussed below (29).

(iii) $\bar{M}_q = 0$: In the chiral limit the flow hits a singularity at a finite $k_\chi > 0$.

As discussed above, we can clearly distinguish the first two regimes in Figure 6. Interestingly, the unphysical regime with $\bar{M}_{q,\Lambda}$ is reached for rather small initial masses roughly two orders of magnitude below the initial cutoff scale. In QCD the physical UV cutoff scale for the NJL model is approximately 1 GeV and hence the unphysical regime is entered for current quark masses of about 10 MeV. Accordingly, this should allow for a chiral extrapolation as also shown

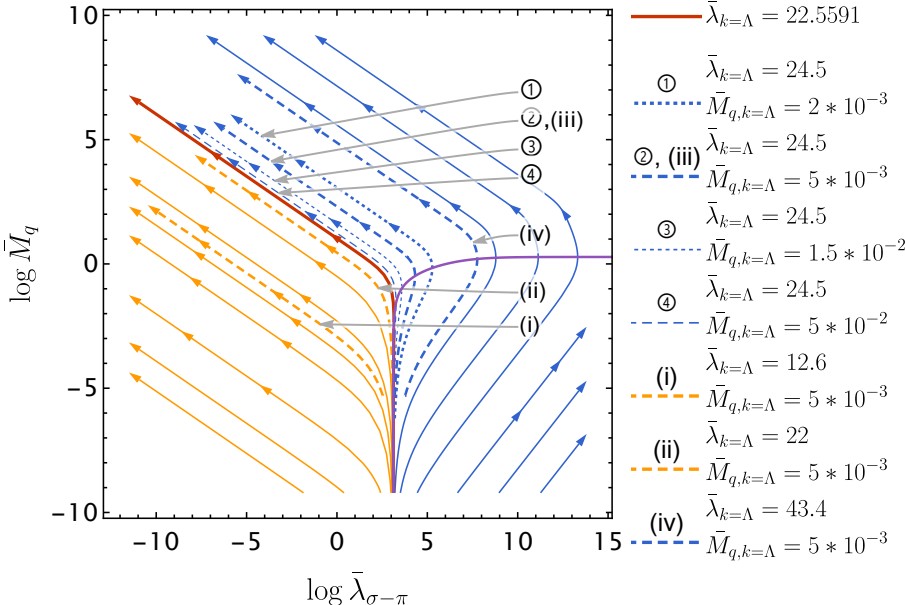

Figure 8: fRG flow diagram in the $\bar{M}_q$-$\bar{\lambda}_{\sigma-\pi}$ plane, with the dimenionless $\bar{M}_q, \bar{\lambda}_{\sigma-\pi}$ given in (18). Arrows indicate the flow towards the infrared. The plane is sepearated into two sub-planes (*blue* and *orange*). *Blue flows* start in the regime with dynamical chiral symmetry breaking with (31) and finally enter the Gaußian regime, *orange flows* are initiated in the Gaußian regime with (32) and stay there. Circled numbers indicate flows with the initial coupling $\lambda_{\sigma-\pi,\Lambda} = 22.5$ and different current quark masses $M_{q,\Lambda}$, see Figure 7. Roman numbers indicate flows with initial current quark mass $M_{q,\Lambda} = 5 \times 10^{-3}$ and different initial couplings $\lambda_{\sigma-\pi,\Lambda}$, see Figure 9.

in Figure 6 with a constituent quark mass of $M_{q,\chi} = 0.242\,\Lambda$ in the chiral limit. This concludes our structural discussion of the local NJL-type flows.

With this preparation we proceed with the discussion the general solutions of the coupled system of flow equation in (20a) and (20b). They are depicted in Figure 8, which can be understood in terms of the discussions in the following:

To begin with, the $\bar{\lambda}_{\sigma-\pi}$-$\bar{M}_q$ plane in Figure 8 is separated into two sub-planes by the *red solid* line, the *orange* and *blue* regimes, respectively.

These two regimes are qualitatively different as only the blue regime carries the dynamics of chiral symmetry breaking: It is determined by all flows which are initiated with

$$\beta_{\bar{\lambda}_{\sigma-\pi}}(\bar{M}_{q,\Lambda}, \bar{\lambda}_{\sigma-\pi,\Lambda}) < 0 \,, \tag{31}$$

and hence the dimensionless coupling $\bar{\lambda}_{\sigma-\pi}$ increases first which indicates the dynamics of spontaneous chiral symmetry breaking. However, for some small $k$ the $\beta$-function vanishes as discussed before. In Figure 8 the violet solid line shows the vanishing $\beta$-function curve (23). Note that the asymptotic line of the violet curve at large $\bar{\lambda}_{\sigma-\pi}$ is given by the line of $\bar{M}_q = \sqrt{7}/2$ as shown in (24). Then, the flow enters the attraction regime of the Gaußian fixed point and the $\bar{\lambda}_{\sigma-\pi}$ decreases towards zero. We call this regime that with *dynamical chiral symmetry breaking*. It is determined by all flows that intersect the violet curve.

In turn, the flows in the *orange* regime are initiated in the attraction regime of the Gaußian fixed point with

$$\beta_{\bar{\lambda}_{\sigma-\pi}}(\bar{M}_{q,\Lambda}, \bar{\lambda}_{\sigma-\pi,\Lambda}) > 0 \,, \tag{32}$$

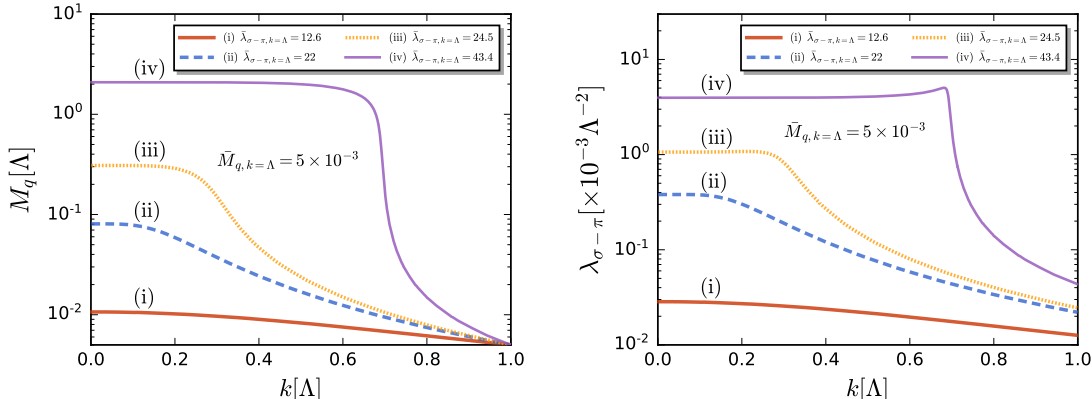

Figure 9: Quark mass $M_q$ (left panel) and four-quark coupling $\lambda_{\sigma-\pi}$ (right panel) as functions of the RG scale $k$ for several different initial values of the coupling $\bar{\lambda}_{\sigma-\pi,k=\Lambda}$ and a fixed initial current quark mass $\bar{M}_{q,k=\Lambda} = 5 \times 10^{-3}$, also shown in Figure 8.

Then, the $\beta$-function is positive for all $k$ and hence the flow stays in the attraction regime of the Gaußian fixed point. We call this regime that with *Gaußian regime*.

The red solid line that separates the two regimes, is given by the flow initiated with the initial conditions as follows

$$\bar{M}_{q,\Lambda} \to 0^+\,, \qquad \bar{\lambda}_{\sigma-\pi,\Lambda} = \bar{\lambda}^*_{\sigma-\pi}(0)\,, \tag{33}$$

which is approaching the boundary from within the Gaußian regime.

We proceed with a discussion of the properties of the two regimes with and without dynamical chiral symmetry breaking.

### 3.1.1 Gaußian regime

Flows in the Gaußian regime are depicted in Figure 8 and Figure 9 with (i) and (ii). These flows are dominated by its dimension counting part $2\bar{\lambda}_{\sigma-\pi}$. The dimensionless coupling is decreasing monotonously as is also visible in Figure 8. Finally it tends towards zero for $k=0$. In turn, the dimensionful coupling is increasing for large $k$ as $B(\bar{M}_q) > 0$ for sufficiently small initial $\bar{M}_q$, and settles at a finite value for $k \to 0$. Note, that $B(\bar{M}_q) < 0$ entails that the initial quark mass is larger than the initial cutoff scale, see (24), in which case all flows are already suppressed at the initial scale.

An interesting detail of the flows in the Gaußian regime is the potentially significant increase of the mass function $M_{q,k}$ towards the infrared. We emphasise that this simply reflects a trivial RG-running in the Gaußian regime. This regime does not sustain the dynamics of spontaneous chiral symmetry breaking. Accordingly, while such a flow leads to larger infrared masses, and this may be used to mimic a constituent quark mass it lacks the respective dynamics.

In short, such a setup can be used to emulate a large constituent quark mass but fails qualitatively to incorporate the respective dynamics. The latter fact then shows in other observables.

### 3.1.2 Regime with dynamical chiral symmetry breaking

Flows in the regime with dynamical chiral symmetry breaking are depicted in Figure 8 and Figure 9 with ①—④ with the initial coupling $\bar{\lambda}_{\sigma-\pi,\Lambda} = 24.5$ and (iv) with the initial coupling $\bar{\lambda}_{\sigma-\pi,\Lambda} = 43.4$.

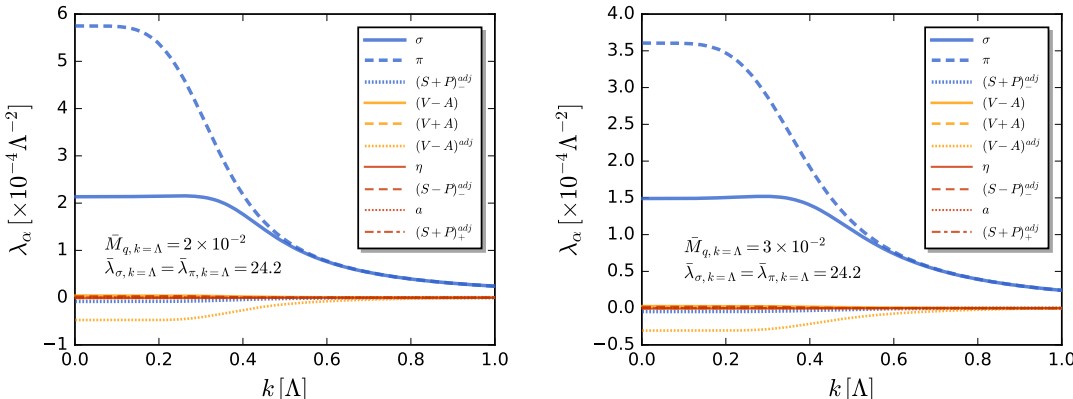

Figure 10: Four-quark couplings $\lambda_{\alpha,k}$ of 10 Fierz-complete channels as functions of the RG scale $k$. In the calculations the initial values of couplings at the UV cutoff $\Lambda$ are chosen to be $\bar\lambda_{\pi,\Lambda} = \bar\lambda_{\sigma,\Lambda} = 24.2$ and $\bar\lambda_{\alpha,\Lambda} = 0$ ($\alpha \notin \{\sigma, \pi\}$) for other channels. Two different initial values of quark mass at $k = \Lambda$ are adopted with $\bar{M}_{q,\Lambda} = 2 \times 10^{-2}$ (left panel) and $3 \times 10^{-2}$ (right panel).

Consequently, $\bar\lambda_{\sigma-\pi}$ is first increasing during the flow towards the infrared as does $\bar{M}_q$, see Figure 8. With the growth of the latter, the system enters the attraction regime $\beta_{\bar\lambda_{\sigma-\pi}} > 0$ of the Gaußian fixed point for small enough $k$, and $\bar\lambda_{\sigma-\pi}$ starts decreasing towards the infrared. This turning point is given by the violet curve of fixed points in Figure 8. This qualitative structure is present for all flows in the blue regime with dynamical chiral symmetry breaking.

We emphasise that while the flows of dimensionful couplings and quark masses do not differ qualitatively from that in Gaußian regime in particular for large initial quark masses, the dynamics does.

Finally, as discussed before, for small initial masses $\bar{M}_{q,\Lambda} \lesssim \bar{M}_\chi$ defined in (29), we detect a non-monotonous dependence of the constituent quark mass $M_q$ on the input (current) quark mass $M_{q,\Lambda}$, see Figure 6 and Figure 7. We remark that this artificial behavior disappears in an improved truncation with a quark propagator with full momentum dependence and momentum-dependent four-quark couplings. A detailed discussion goes beyond the scope of the present work and is deferred to [40].

## 3.2 Fierz complete approximation

We close this section with presenting the results in the best approximation considered here: We use a Fierz-complete basis of the four-quark interaction with the ten momentum-independent tensor structures, see (A.1) and (16). At the initial scale $k = \Lambda$ we only keep a non-vanishing scalar-pseudoscalar coupling $\bar\lambda_{\sigma-\pi}$, all the other couplings are set to zero,

$$\bar\lambda_{\sigma,\Lambda} = \bar\lambda_{\pi,\Lambda} = 24.2, \qquad \bar\lambda_{\alpha \notin \{\sigma,\pi\},\Lambda} = 0. \tag{34a}$$

Identifying the scalar and pseudoscalar couplings at the initial scale assumes a negligible effect from the explicit chiral symmetry breaking due to the initial (current) quark mass $\bar{M}_{q,\Lambda}$. The latter is kept small in units of the initial cutoff, and varied within one order of magnitude $5 \times 10^{-3} - 5 \times 10^{-2}$ and is used change the initial conditions from the *orange* Gaußian regime in Figure 8 to the *blue* regime with dynamical chiral symmetry breaking,

$$\bar{M}_{q,\Lambda} = 5 \times 10^{-3}, \quad (1, 2, 3, 4, 5) \times 10^{-2}. \tag{34b}$$

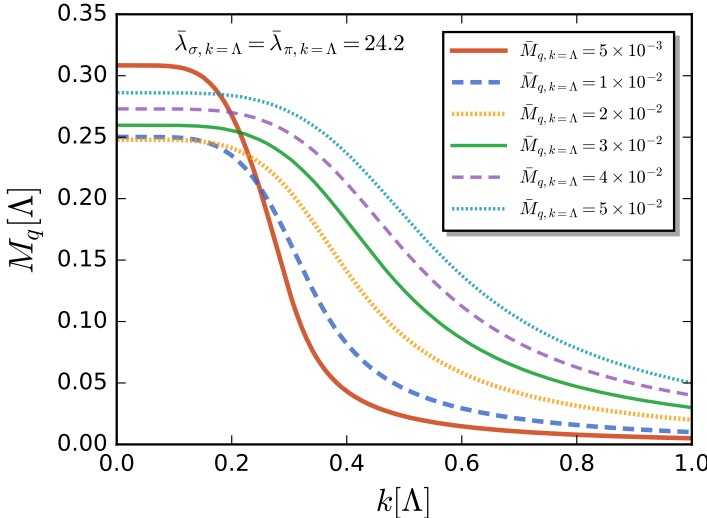

Figure 11: Quark mass as a function of the RG scale $k$ obtained in the computation with four-quark interactions of 10 Fierz-complete channels. Results for several different initial values of $\bar{M}_{q,k=\Lambda}$ are compared. The initial values of four-quark couplings are fixed with $\bar{\lambda}_{\pi,k=\Lambda} = \bar{\lambda}_{\sigma,k} = 24.2$ and $\bar{\lambda}_{\alpha\notin\{\sigma,\pi\},\Lambda}$.

This setup allows us also to evaluate the relevance of the additional channels by successively switching them off and comparing the respective results to that of the full Fierz-complete computation. Finally, we estimate the quantitative reliability of the simple one-channel approximation as discussed in Section 3.1.

The results of the Fierz-complete computation are presented in Figure 10 and Figure 11. In Figure 10 we depict the coupling strengths of different channels as functions of the cutoff scale $k$. As expected, the $\sigma$ and $\pi$ channels are dominant for all $k$. Moreover, for the initial masses in (34b) they stay degenerate for $k \gtrsim 0.6\Lambda$ as can be seen for $\bar{M}_{q,\Lambda} = 3 \times 10^{-3}$ in Figure 10. For even larger masses this regime shrinks towards $k = \Lambda$ and finally the assumed degeneracy is not self-consistent anymore. However, for the present masses the degeneracy as used in (34a) is supported well.

In turn, for even smaller cutoff scales, the two channels are not degenerate anymore, which signals the onset of dynamical chiral symmetry breaking. Then, $\lambda_\pi$ is significantly larger than $\lambda_\sigma$ due to the presence of pseudo-Goldstone modes, the pions, see Figure 10. This already indicates, that the single-channel approximation used in Section 3.1 may not provide quantitative precision. Indeed, one finds that the results agree only qualitatively.

In Figure 11 we show the dependence of the quark mass on $k$ with several values of $\bar{M}_{q,\Lambda}$ for the Fierz complete approximation. The results agree qualitatively but not quantitatively with the results in the left panel of Figure 7. As the running quark mass is sensitive to the accumulated fluctuations in all channels, it is an optimal quantity for evaluating the convergence (or deviation) of results of different approximations towards results in the Fierz-complete one.

The large dominance of the scalar-pseudoscalar channel and in particular that of the pion exchange with $\lambda_\pi$ suggests that the feedback of the other channels in the flow is negligible. This can be tested with switching off $\lambda_{\alpha\notin(\sigma,\pi)}$ on the right hand side of the flow equations and flowing down from the same initial conditions (34). We find that this changes the results just by a few percent: less than 2% for the constituent quark mass, less than 3% for $\lambda_\sigma$ and less than 5% for $\lambda_\pi$, see also Figure 16 and Figure 17 in Appendix D. In this approximation the couplings $\lambda_{\alpha\notin\{\sigma,\pi\}}$ are simply generated from $\lambda_\sigma, \lambda_\pi$ diagrammatically. They deviate from the full results

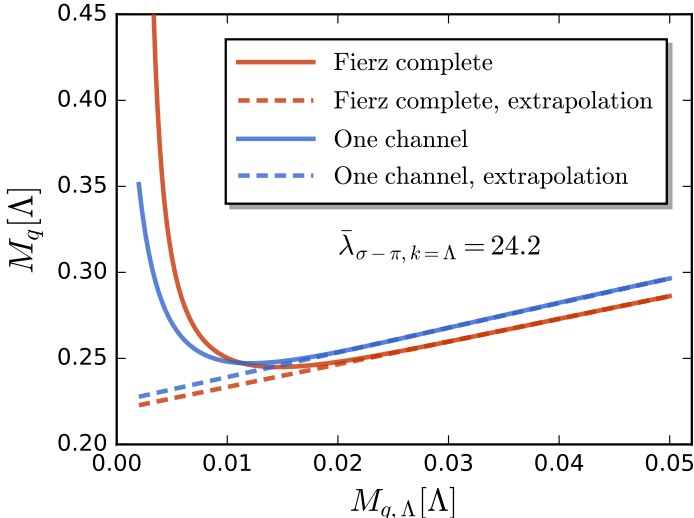

Figure 12: Physical constituent quark mass $M_{q,k\to 0}$ as a function of the UV (current) quark mass $M_{q,\Lambda}$ obtained from the Fierz-complete (red solid) and the one-channel (blue solid) computations, both with an initial coupling $\bar{\lambda}_{\sigma-\pi,\Lambda} = 24.2$, cf. also Figure 6 for the one-channel computation. The respective linear extrapolations towards the chiral limit are depicted as dashed lines, and is supported by its presence within dynamical hadronisation, e.g. [54].

by less than 2%, see Figure 18 in Appendix D. This confirms the quantitative dominance of the $\sigma-\pi$ quantum fluctuations in this system. These findings corroborate results in the literature within Fierz complete systems both in NJL-type models and in QCD [18, 20, 29–31].

While being subdominant, we also find an ordering in the strength of the couplings $\lambda_{\alpha\notin(\sigma,\pi)}$: In the present tensor basis with (A.1), the strengths of the $(V-A)^{\text{adj}}$ and $(S+P)^{\text{adj}}_-$ channels significantly exceed that of the further channels. A similar pattern has also been seen in other Fierz complete computations in both NJL-type models, see e.g. [29, 30], and QCD [18, 20, 31]. A direct comparison of the results in the present work with those in the literature is not possible: in [29–31] the computations are performed in the chiral limit and hence the flows stop at $k_\chi$, where the flow hits a singularity. The flows in [18, 20] are computed in full QCD and use a slightly different basis. We have checked in all cases, that our results are compatible with those in the literature, if these differences are taken into account.

We close our analysis of the dynamics of spontaneous chiral symmetry breaking with a comparison of the approach towards the chiral limit as already discussed for the single-channel approximation. There we have discussed the constituent quark masses $M_q$ as a function of the current quark mass $\bar{M}_{q,\Lambda}$, see depicted Figure 6. In Figure 12 we compare $M_q(M_{q,\Lambda})$ for the single channel approximation (*blue curve*) with the results of the Fierz-complete one (*red curve*). We have already discussed, that the results from the two approximations do not agree quantitatively and we have to tune both initial coupling for the best comparison: in Figure 12 we use $\bar{\lambda}_{\sigma-\pi,\Lambda} = 24.2$ for the Fierz complete results (*red curve*), and $\bar{\lambda}_{\sigma-\pi,\Lambda} = 24.5$ as already used for Figure 6 for the single-channel approximation.

In the linear regime of both curves for $M_{q,\Lambda} \gtrsim 0.02$, the constituent masses (and their slopes) on both curves deviate less than 5%. Since the initial current quark masses are less than 20% of the final constituent quark masses, this entails that the pseudoscalar-scalar channel is dominating completely the dynamics in the linear regime. This is also seen from comparing the one-channel results for $\bar{\lambda}_{\sigma-\pi,\Lambda} = 24.2$ in Figure 12 and $\bar{\lambda}_{\sigma-\pi,\Lambda} = 24.5$ in Figure 6. Note

that this 1% change of the coupling already gives rise to a larger change ($\approx$ 10%) of the constituent quark masses in comparison to adding all channels (less than 2%).

In turn, for small current quark masses with $M_{q,\Lambda} \lesssim 0.02$ we enter the unphysical non-linear regime where the four-quark couplings are oversized. We expect that the related unphysical infrared divergence of the constituent quark mass is amplified by the presence of all channels. Indeed the Fierz complete computation (*red curve*) deviates earlier and stronger from the linear behaviour.

# 4 Emergent bound states

We continue with the main goal of the present work, the evaluation of emergent hadronic bound states within the present formulation.

So far, respective fRG computations in QCD have been done within the framework of dynamical hadronisation [5, 17, 18, 20–22, 24, 55], see also the reviews [43, 44]. In short, in this framework composite bilinear fields $\bar{q}\mathcal{T}q$ are introduced, that comprises the full dynamics of one momentum channel of a four-quark exchange $(\bar{q}\mathcal{T}q)^2$ with the given tensor structure $\mathcal{T}$. We emphasise that dynamical hadronisation only *reparametrises* the QCD effective action, it does not suffer from double counting problems, well-known from standard Hubbard-Stratonovich transformations.

In most cases dynamical hadronisation has been used for the $t$-channel of the scalar-pseudoscalar tensor structure. The respective composite fields carry the same quantum numbers as the pseudoscalar mesons, the pions, and the scalar $\sigma$-mode. More importantly, they carry the pole masses and decay properties of the respective mesons. Finally, higher order scatterings of resonant interaction channels (such as multi-pion scatterings) are conveniently and reliably accounted for with a full effective potential of the composite fields. This is chiefly important in the presence of quasi-massless modes as well as phase transitions.

In such a formulation all the other tensor channels of the four-quark interactions as well as a remnant momentum dependence have to be treated independently: This can be done either within further dynamical hadronisation steps or by keeping the rest of the four-quark interactions explicitly in the system. This is discussed in detail in [22].

Hence, a systematic and reliable treatment of the bound state spectrum of QCD as well as the phase structure of QCD at larger densities with the fRG asks for a resolution of the dynamics of resonant interaction channels as well as an approximation in the four-quark scatterings or even higher order quark interactions, that can capture the emergence of resonant channels.

In the present Section we initiate the analysis of reliable approximations in the four-quark sector of QCD with a detailed study of the dynamics of the scalar-pseudoscalar channel. In consequence of the findings of the last sector we shall use a $t$-channel approximation of the scalar-pseudoscalar four-quark scatterings. This is similar to dynamical hadronisation with a full momentum dependence of the pion and $\sigma$ propagators and a momentum-independent Yukawa coupling between $(\sigma, \pi)$ and $(\bar{q}q, i\bar{q}\gamma_5\tau q)$. This approximation is well-known for capturing the infrared dynamics of pions and $\sigma$ well, and we confirm this here without dynamical hadronisation. This shows the capability of the present framework to account for potentially resonant interaction channels.

We also compute the timelike regime of the $t$-channel which gives us access to the pion pole mass as well as further scattering properties such as the pion decay constant. A detailed evaluation of the on-shell properties of the pions and $\sigma$ is beyond the scope of the present work and will be discussed elsewhere. Finally, we complete the present case study with a discussion of the regulator dependence or rather lack thereof.

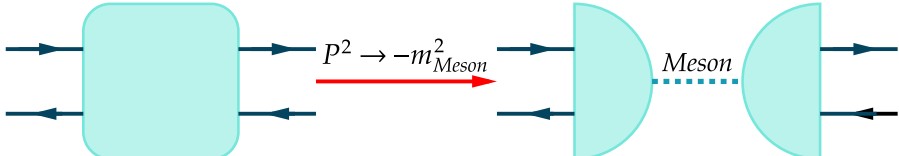

Figure 13: Sketch of four-quark vertex and it's resonance behavior at the pole of relevant meson mass. Here the square and half-circles denote the full four-quark and quark-meson vertices, respectively. The dashed line stands for the meson propagator.

## 4.1 Pion pole mass and the Goldstone theorem

The information of bound states of, e.g., two quarks (or one quark and one antiquark), or three quarks are encoded in the four-point or six-point vertices of quarks, respectively, see [3] for the DSE-BSE approach and [22] for the fRG approach. This is schematically depicted in Figure 13, where we have taken mesons as a simple example: We consider Euclidean momenta $P$ of the respective tensor channel (e.g. the pseudoscalar channel for the pion) with $P^2 \to -m_{\text{meson}}^2$, that is close to on-shell momenta. Then the respective tensor channel in the four-quark vertex is resonant, and the full four-quark vertex can be well approximated as two quark-meson vertices connected with a meson propagator, as shown in the right part of Figure 13. Accordingly, we have to access the full four-quark vertex or the quark-meson vertex in this momentum regime.

### 4.1.1 Flow equations in the $t$-channel approximation

The resummation required for the four-quark couplings in this momentum channel is already encoded in the flow equation in (B.1c). For our pion example it is $\lambda_\pi(P)$. Naturally, the fRG flows are therefore well-suited for the description of bound states. Properties of the meson in a given channel can be inferred from its respective four-quark coupling in (B.1c) with an appropriate external momentum for the meson. We emphasise that the RG flows in (B.1c) and (B.1b) have a welcome property, that *any* truncation to the effective action in (2) leads to self-consistency of the quark propagator and the emerging bound states. In turn, the self-consistency between gap equation and the Bethe-Salpeter equation is a nontrivial requirement [3].

We proceed with investigating the pole behavior of mesons as shown in Figure 13 with the flow equations of four-quark couplings in (B.1c). It is obvious that the truncation for the coupling in (17), where all the external momenta are ignored, is not applicable any more for the current purpose, but rather partial momentum dependence at least should be restored.

The relevant Mandelstam variables read

$$s = (p_1 + p_2)^2,$$

$$t = (p_1 + p_3)^2 = P^2,$$

$$u = (p_1 + p_4)^2. \tag{35}$$

For momenta close to the on-shell condition of the resonance, this bound state naturally emerges from the flow equation of its relevant four-quark coupling in (B.1c), see Figure 13. In this work we are interested in the qualitative dynamics of such a system in the four-quark flows in (B.1c). Hence we only keep the momentum dependence of the resonant channel. For the $\pi$ meson under investigation we use the $t$-channel in (35) with $P^2 \sim -m_\pi^2$, while keeping

$s = u = 0$, that is $p_1 = -p_2 = -p_4$. Thus, one is led to

$$t = P^2 \to -m_\pi^2, \qquad s = u \to 0. \tag{36a}$$

For these momenta the pion coupling $\lambda_\pi$ is divergent and all other four-quark couplings are negligible: As the system is coupled and the couplings are not orthogonal, they also diverge but $\lambda_\pi$ is by far dominant. Moreover, its $t$-channel dominates over the rest of the momentum dependence as indicated in Figure 13.

In the present work we concentrate on the scalar-pseudoscalar $t$-channel vertex and evaluate its full flow in Appendix B on the $t$-channel configuration (36a) with

$$p_1 = p_3 = -P/2. \tag{36b}$$

The diagrams are depicted in the second line of Figure 4. For the momentum configuration (36a) with (36b) only the first diagram on the r.h.s. of the flow equation for the four-quark functions has vertices with a momentum flowing through the diagram, and this momentum is simply the $t$-channel momentum $P$. Moreover, its vertices are of the form,

$$\Gamma_{\bar{q}_1 \bar{q}_i q_3 q_j}(-P/2, -q, -P/2, P+q) \propto \lambda(P),$$

$$\Gamma_{\bar{q}_{j'} \bar{q}_2 q_{i'} q_4}(-P-q, P/2, q, P/2) \propto \lambda(P), \tag{36c}$$

where all momenta are counted inflowing and $i, j, i', j'$ labels the internal lines. In (36c) we have also used that the loop momenta $q$ are limited by the cutoff scale, $q^2 \lesssim k^2$, and the momentum dependence of the vertices is subleading for these momenta: The initial vertices are momentum independence and hence the momentum dependence originates from that of the propagators. For $q^2 \lesssim k^2$ the propagators are dominated by the regulator $R_q^2(q^2 \lesssim k^2) \sim k^2$ which is approximately constant in this regime. Accordingly, for $P^2 \gg k^2$ these vertices are well approximated by those with $q = 0$, leading to (36c). The validity of this conceptual argument has been tested with the full results in [18, 20] in QCD.

Importantly, the approximation (36c) leads to a factorisation in the first diagram on the r.h.s. of the flow equation in the second line in Figure 4: The vertex dressing $\lambda(P)$ multiplies the loop, whose $P$-dependence only comes from the propagators.

The above argument also implies that the vertices in the second and third diagrams are well approximated by their values at vanishing momenta, e.g.,

$$\Gamma_{\bar{q}\bar{q}_i q q_j}(P/2, -q, -P/2, q) \propto \lambda(0). \tag{36d}$$

Finally, the pion channel $\lambda_\pi$ has the dominant resonance, and hence we can use

$$\lambda_\alpha(P) \to \lambda_\alpha(0), \qquad \alpha \neq \pi, \tag{36e}$$

in the resonant first diagram in Figure 4. In summary, it is this diagram which gives rise to the dominant momentum dependence in the flow, and hence the (integrated) flow equation for $\lambda_\pi(P)$ will relate to standard $t$-channel resummations.

Within the $t$-channel approximation described above and summarised in (36), we are led to the flow

$$\partial_t \lambda_\pi(P) = \mathcal{A}(P) + \mathcal{B}(P)\lambda_\pi(P) + \mathcal{C}(P)\lambda_\pi^2(P), \tag{37}$$

where we have suppressed the $k$-dependence for the sake of simplicity. The two terms proportional to $\lambda_\pi(P)$ and $\lambda_\pi^2(P)$ with the $k$-dependent coefficients $\mathcal{B}(P)$, $\mathcal{C}(P)$ are the scalar-pseudoscalar contribution of the first diagram on the r.h.s. in the second line in Figure 4.

The factorisation described above is explicit in these terms. Note that $\mathcal{B}(P)$ is linear in the other couplings $\lambda_{\alpha\neq\pi}$. In present $t$-channel approximation all coefficients have simple representations in terms of loop diagrams with constant vertex factors and momentum dependent propagators. They can be read off from the general flows in Appendix B and Appendix C, which are collected in Appendix E.

The three terms in (37) are ordered in increasing powers of the resonant coupling $\lambda_\pi$ and hence are increasingly important in the resonant regime. Their coefficients have a natural interpretation in terms of the $\beta$-function of $\lambda_\pi$ depicted in Figure 5:

The first coefficient $\mathcal{A}$ leads to a global shift of the $\beta$-function up or down depending on its sign. This leads to a shifted Gaußian infrared fixed point as well as a shift of the UV fixed point $\lambda_\pi^*$ to larger values ($\text{sign}(\mathcal{A}) > 0$) or smaller values ($\text{sign}(\mathcal{A}) < 0$). In QCD this term also carries negative gluon contributions proportional to $\alpha_s^2$ and beyond a critical coupling $\alpha_s > \alpha_s^*$, the UV fixed point disappears and the $\beta$-function is negative for all values of $\lambda_\pi$.

The second coefficient linear in $\lambda_{\alpha\neq\pi}$ accounts for the anomalous dimension of the vertex dressing and shifts the canonical term $2\bar{\lambda}_\pi \to (2 + \mathcal{B})\bar{\lambda}_\pi$. In QCD it also includes terms linear in $\alpha_s$. In contradistinction it has no qualitative impact for $\mathcal{B} > -2$ and hence is of no importance in our present qualitative analysis.

Finally, the third and most relevant coefficient $\mathcal{C}$ is $\lambda$-independent and is simply the loop integral of the $t$-channel diagram in the second line in Figure 4 with $\lambda_\pi = 1$. As discussed above, its momentum dependence triggers that of $\lambda_\pi(P)$ in the present approximation.

### 4.1.2 Persistence of the Goldstone theorem

We are now in the position to discuss the natural emergence of the pion bound state and the persistence of the Goldstone theorem within the current approximation.

In Section 4.2.1 and Section 4.2.2 we numerically compute the coupling $\lambda_{\pi,k}$ from the flow (37) in the following way: While the momentum-dependent vertex dressing $\lambda_\pi(P)$ and the momentum-independent vertex dressings $\lambda_{\alpha\neq\pi}$ are computed from their respective coupled flow equations, the quark propagator is taken as an input and computed within the approximation detailed in Section 3.2.

Accordingly, this approximation is not self-consistent for general momenta $P^2 \neq 0$. However, it is self-consistent for $P = 0$ due to the factorisation discussed before: At $P = 0$ (37) reduces to

$$\partial_t \lambda_\pi(0) = \mathcal{A}(0) + \mathcal{B}(0)\lambda_\pi(0) + \mathcal{C}(0)\lambda_\pi^2(0),\tag{38}$$

which is the approximation used in Section 3.2 for computing the coupling in the coupled system for $M_{q,k}, \lambda_k$. There it has been shown that the coupling only diverges for $M_{q,\Lambda} \to 0$, tantamount to the occurrance of a massless mode in the system. Hence it is precisely the self-consistency for $P = 0$, which leads to the persistence of the Goldstone theorem, see also the explicit results in Section 4.2.1 and Section 4.2.2.

We now elucidate the above with an analytic computation within a simplified approximation that already captures the relevant structure: As the first term $\mathcal{A}_k(P)$ in (37) is regular in the chiral limit as well as more generally for $P^2 \to -m_\pi^2$, we can drop it for a first qualitative study. For $\mathcal{A}_k = 0$, the flow (37) is readily integrated,

$$\lambda_\pi(P) = \frac{\lambda_{\pi,\Lambda}\mathcal{D}_0(P)}{1 - \lambda_{\pi,\Lambda}\int_\Lambda^0 \frac{dk}{k}\mathcal{D}_k(P)\mathcal{C}_k(P)},\tag{39}$$

with

$$\mathcal{D}_k(P) \equiv e^{\int_\Lambda^k \frac{dk'}{k'}\mathcal{B}_{k'}(P)}.\tag{40}$$

Table 2: Pion pole mass with different current quark masses. The results in the first line denote those obtained in the direct computations in the Minkowski regime, while others are obtained from analytic continuation from the Euclidean to Minkowski regimes based on Padé approximants of different orders. The calculations are done with the $3d$ flat regulator, where the initial values of four-quark couplings are fixed with $\bar{\lambda}_{\pi,k=\Lambda} = \bar{\lambda}_{\sigma,k=\Lambda} = 16.92$ and $\bar{\lambda}_{\alpha,k=\Lambda} = 0$ ($\alpha \notin \{\sigma, \pi\}$), see also Figure 14.

| $M_{q,k=\Lambda}[\Lambda]$ | $10^{-4}$ | $10^{-3}$ | $5 \times 10^{-3}$ | $2 \times 10^{-2}$ |
|---|---|---|---|---|
| Exact | 0.03683 | 0.1185 | 0.2067 | 0.3375 |
| Padé $[1,1]$ | 0.07120 | 0.1139 | 0.2039 | 0.3359 |
| Padé $[2,2]$ | 0.03812 | 0.1185 | 0.2068 | 0.3382 |
| Padé $[10,10]$ | 0.03683 | 0.1185 | 0.2067 | 0.3374 |
| Padé $[20,20]$ | 0.03683 | 0.1185 | 0.2075 | 0.3281 |

Here, $\lambda_\pi = \lambda_{\pi,k=0}$ and the initial four-quark coupling $\lambda_{\pi,\Lambda}$ is momentum-independent.

Evidently, the physical four-quark coupling $\lambda_\pi(P)$ diverges, if the denominator in (39) crosses through zero. Therefore, the pole mass of a bound state, here the pion, can be determined through the equation as follows

$$\int_\Lambda^0 \frac{dk}{k} \mathcal{D}_k(P) \mathcal{C}_k(P) = \frac{1}{\lambda_{\pi,\Lambda}}, \tag{41}$$

for $P^2 = -m_\pi^2$. In the presence of explicit chiral symmetry breaking with $m_\pi^2 > 0$ the resolution of the pole condition (41) requires timelike momenta $P^2 < 0$.

In the chiral limit with $M_{q,\Lambda} = 0$, the pion is massless due to the Goldstone theorem, $m_\pi^2 = 0$. Hence, the persistence of the Goldstone theorem in the current approximation implies

$$\int_\Lambda^{k_\chi} \frac{dk}{k} \mathcal{D}_k(0) \mathcal{C}_k(0) + \int_{k_\chi}^0 \frac{dk}{k} \mathcal{D}_k(0) \mathcal{C}_k(0) = \frac{1}{\lambda_{\pi,\Lambda}}, \tag{42}$$

where $k_\chi$ is the chiral symmetry breaking cutoff scale in the chiral limit,

$$M_{q,k} \equiv 0, \qquad \text{for} \qquad k > k_\chi. \tag{43}$$

The first term on the l.h.s. of (42) can be regarded as a constant due to the vanishing quark mass in (43), and the second term only depends on $M_{q,k}$, which itself is a function of $\lambda_{\pi,\Lambda}$.

In conclusion in the present fRG approach the self-consistency of a BSE-DSE system of the BSE system for the four-quark scattering kernel and the DSE gap equation translates into the combined (self-consistent) solution of the flow equation for the quark propagator and the four-quark vertex. It is worth mentioning that within dynamical hadronisation the necessity for a self-consistent flow is absent as all these features are carried by the flow equation of the mesonic potential of pion and sigma mode: Every approximation of the latter carries the Goldstone theorem independent of the approximation of the flow for the quark propagator, for more details see the recent works [22, 43, 44] and references therein.

## 4.2 Numerical results

In this Section we present the results of our explicit computations within the approximation described in detail in Section 4.1. The present fRG approach allows for a direct access to

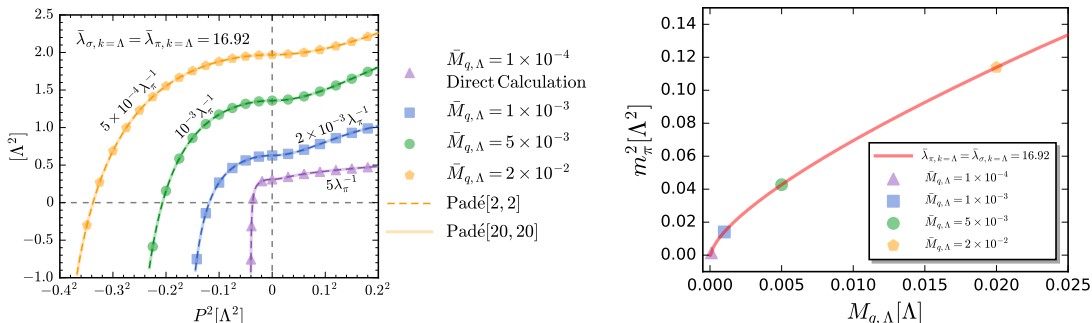

Figure 14: Left panel: Inverse of the four-quark coupling of the $\pi$ channel, $1/\lambda_{\pi,k=0}$, as a function of the Mandelstam variable $t = P^2 = P_0^2 + \boldsymbol{P}^2$ with $\boldsymbol{P} = 0$. Here the $3d$ flat regulator is used. Results for several different initial values of $\bar{M}_{q,k=\Lambda}$ are compared. The initial values of four-quark couplings are fixed with (46a). Data points denote the results calculated directly in the analytic flow equation as shown in (37) both in the Euclidean ($P^2 > 0$) and Minkowski ($P^2 < 0$) regions. The dashed and solid lines stand for results of analytic continuation from $P^2 > 0$ to $P^2 < 0$ based on the fit of the Padé [2,2] and Padé [20,20] approximation, respectively.
Right panel: Squared pion mass as a function of the initial value of the quark mass $M_{q,\Lambda}$ obtained with the $3d$ regulator. The same initial values of four-quark couplings are employed as the left panel. The several different points denote the four values of $m_\pi^2$ corresponding to those extracted from the left panel.

timelike momenta $P^2 < 0$ for appropriate regulators: The respective properties are discussed in detail in [56].

In short, if one wants to maintain a spectral representation and the standard Wick rotation as well as Lorentz invariance for finite cutoff scales, one is left with the masslike Callan-Symanzik regulators which explicitly breaks chiral symmetry. Then, chiral symmetry is restored for $k \to 0$ and this restoration can be monitored with modified symmetry identities. This is a viable option but we do not want to burden the present work with yet another layer of complexity and we restrict ourselves to chirally symmetric regulators.

Chiral symmetry is maintained with the class of spatial momentum regulators with Dirac structure. This choice is used in Section 4.2.1, see (44). While this choice breaks the Euclidean $O(4)$ symmetry and hence also Lorentz symmetry, it allows us to directly compute the four-quark dressings at timelike momenta even for finite cutoff scales. For $k \to 0$ full Euclidean and Lorentz symmetry is restored. The direct computational results for timelike momenta also give us access to the highly interesting question of the validity of spectral reconstructions.

Another chirally invariant regulator class is that with full Euclidean $O(4)$ symmetry and hence full Lorentz symmetry. This choice is used in Section 4.2.2, see (50). However, all known classes of regulators generate additional poles or cuts in the complex frequency [56]. These additional singularities also complicate the direct evaluation at timelike momenta at $k \neq 0$. Indeed, in the case of the four-dimensional flat regulator the theta function is even non-analytic from the onset and has no natural extension to $P^2 < 0$. However, for $k \to 0$ these additional singularities are removed. Therefore we can apply standard reconstruction methods for our results at $k = 0$ on the basis of the consistency checks for reconstructions done for the results with the spatial momentum regulators.

### 4.2.1 Results with spatial momentum regulators

Here we present results with an flat spatial momentum regulator,

$$R_{\bar{q}q} = Z_q \mathrm{i} \not{p}\, r_q(\boldsymbol{p}^2/k^2), \tag{44}$$

with

$$r_q^{\text{flat}}(x) = \left(\frac{1}{\sqrt{x}} - 1\right)\Theta(1-x), \tag{45}$$

see also (G.1) in Appendix G. Within this choice the loop momentum integrals can be computed analytically as long as classical dispersions are used in the propagators. Within the current approximation with classical scale-dependent quark dispersions this allows for some analytic cross-checks.

We emphasise that for more advanced approximations as also used in [40], all computations are purely numerical. Then smooth regulators such as the exponential one, see (G.2) in Appendix G are advantageous: To begin with, functional optimisation entails that the flat regulator is only optimal within the local potential approximation. Optimisation in the presence of scale-dependent wave function renormalisations already leads to smooth analytic regulator, [25]. Moreover, these regulators with shape functions such as (G.2) are then taken for computational convenience: The shape function decays rapidly for large spatial momenta and hence facilitates the computation of the loop integrals, as is the fact that it is infinitely differentiable. The lack of the latter property is a numerical obstruction for the use of non-analytic regulators such as the flat regulator (45). In any case, while the present results are obtained within the specific choice (44). They persist also for other regulator shape functions $r_q(\boldsymbol{p}^2/k^2)$, and results for the exponential spatial momentum regulator are discussed in Appendix G.

As discussed above, the spatial momentum regulators allows us to do the calculations directly in the region of $P^2 < 0$. The flow equation in (37) is solved with the $3d$ regulators and four-quark interactions of all 10 Fierz-complete channels with the initial conditions

$$\bar{\lambda}_{\pi,\Lambda}(P) = \bar{\lambda}_{\sigma,\Lambda} = 16.92, \qquad \bar{\lambda}_{\alpha \notin \{\sigma,\pi\},\Lambda} = 0, \tag{46a}$$

in the chirally symmetric phase. Note that the symmetry $\bar{\lambda}_\pi = \bar{\lambda}_\sigma$ is lost for large initial current quark masses $M_{q,\Lambda}$. We choose $\bar{M}_{q,\Lambda} = M_{q,\Lambda}/\Lambda$ as

$$\bar{M}_{q,\Lambda} = 10^{-4}, \, 10^{-3}, \, 5 \times 10^{-3}, \, 2 \times 10^{-2}, \tag{46b}$$

far lower than the initial cutoff scale $k = \Lambda$. Accordingly, the approximation $\bar{\lambda}_{\pi,\Lambda} \approx \bar{\lambda}_{\sigma,\Lambda}$ used in (46a) is well justified. Indeed, the difference between $\bar{\lambda}_{\pi,k}$ and $\bar{\lambda}_{\sigma,k}$ develops safely below $k = \Lambda$, see Figure 10.

As discussed in Section 4.1.1 and indicated in (46a), we only take the dominant $\pi$ channel coupling $\lambda_{\pi,k}(P)$ momentum-dependent, while the other channel couplings $\lambda_{\alpha \neq \pi,k}$ are momentum independent.

The results for the inverse physical channel dressing $1/\lambda_\pi(P) = 1/\lambda_{\pi,k=0}(P)$ are depicted in the left panel of Figure 14: Different colors and shapes of the data points refer to different initial values of the quark mass at the UV cutoff $k = \Lambda$, (46b). While we have full Euclidean and Lorentz symmetry at $k = 0$, the spatial regulator breaks it for $k \neq 0$. Therefore we take $P = (P_0, 0)$ which should minimise any remnant breaking effect due to the cutoff integration. Moreover, the use of spatial momentum regulators allows us to compute the flow equation of the four-quark coupling both in the Euclidean ($P^2 > 0$) and Minkowski ($P^2 < 0$) regimes. The pion pole mass $m_\pi^2$ is determined by

$$\frac{1}{\lambda_\pi(P_0^2 = -m_\pi^2)} = 0, \tag{47}$$

the position of the pole of $\lambda_\pi(P)$. In Figure 14 (left panel) this is simply the intersection point of the curves $1/\lambda_{\pi,k=0}$ with the horizontal dashed line.

The direct Minkowski results also allow us to test the validity of reconstruction methods for the pole position based on Euclidean data. As we are only interested in the location of the first pole or cut in the complex plane, we may use a Padé approximant for the four-quark coupling in term of the Mandelstam variable $t = P^2 > 0$ in the Euclidean region,

$$\lambda_{\pi,k=0}(P^2) \approx \lambda_{\pi,k=0}[n,n](P^2) \equiv \frac{1 + \sum_{i=1}^n a_i (P^2)^i}{\sum_{i=0}^n b_i (P^2)^i}, \tag{48}$$

and the data are taken from the regime

$$P^2 \in [0, (0.2\Lambda)^2]. \tag{49}$$

The timelike momentum results are then readily obtained from (48), evaluated for $P^2 < 0$. We have chosen diagonal Padé fractions indicated by $[n,n]$ as the coupling does not show any decay behaviour in the regimes (49). The order of the Padé approximants employed is varied with $n \in [1,20]$, and some selective results are presented in Table 2 and in the left panel of Figure 14, where the exact results calculated directly in Minkowski regime are also presented for comparison. Note that in Table 2 errors from fitting are not shown, since they are very small for a fixed $n$, especially when $n$ is large. In the left panel of Figure 14 the analytically continued results are depicted as dashed lines (Padé $[2,2]$) and solid lines (Padé $[20,20]$). One can see that when the order is $n \geq 2$, the analytically continued results agree quantitatively with those from the direct computations. The lowest order of $n = 1$ is not adequate, and there are some sizable distinctions, in particular for small current quark masses.

In the right panel of Figure 14 we show the dependence of the extracted squared pion mass on the quark mass at the UV cutoff $k = \Lambda$, i.e., the current quark mass. It is found that the dependence is well approximated as $m_\pi^2 \sim M_{q,\Lambda}$, i.e., $m_\pi^2$ is linearly proportional to $M_{q,\Lambda}$, with $\bar{M}_{q,\Lambda} \gtrsim (2 \sim 3) \times 10^{-3}$. This is consistent with the Gell-Mann–Oakes–Renner relation [57]. However, the linear relation is violated when $\bar{M}_{q,\Lambda}$ is very small, which is attributed to the fact that the flows of the quark mass and the four-quark couplings are not solved self-consistently in this work. In our future work in [40], the self-consistency between the quark propagator and the four-quark couplings are obtained.

In short, we have shown that the present fermionic fRG approach can be used reliably to compute bound state properties of hadrons.

### 4.2.2 Results for regulators with Lorentz symmetry

The same calculations for the momentum-dependent four-quark coupling in (37) are also done with the $4d$ regulator,

$$R_{\bar{q}q} = Z_q \mathrm{i}\slashed{p}\, r_q(p^2/k^2), \tag{50}$$

with the flat shape function (45). As discussed in Section 4.2.1, for more advanced approximations smooth regulators are more convenient both computationally as well as in the context of optimised fRG flows. For comparison we also present results obtained with an exponential shape function (G.2) in Appendix G.

The results on the (inverse) four-quark coupling $1/\lambda_\pi(P)$ are depicted in Figure 15.

In contradistinction to the spatial momentum regulators a direct computation at timelike momenta with $P^2 < 0$ is qualitatively more difficult. However, we have already argued in Section 4.2.1 that the location of the pion pole can be extracted quantitatively by using simple Padé approximants. This has also been confirmed by a comparison of the direct computation to

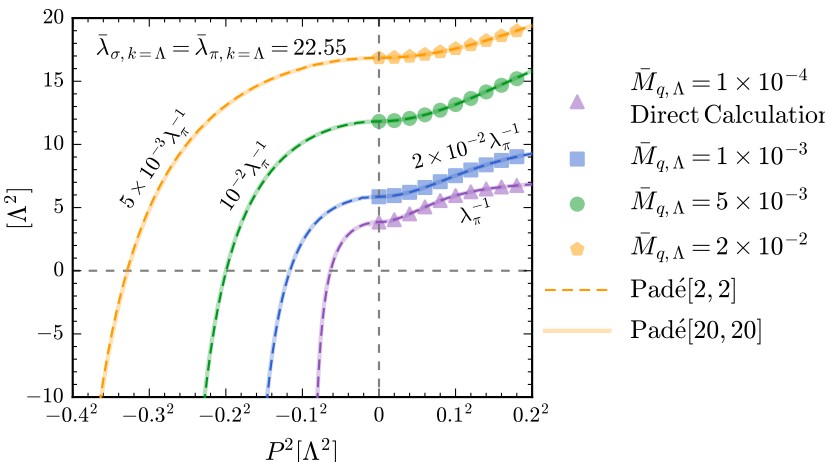

Figure 15: Inverse four-quark coupling of the $\pi$ channel, $1/\lambda_{\pi,k=0}$, as a function of the Mandelstam variable $t = P^2 = P_0^2 + \boldsymbol{P}^2$ with $\boldsymbol{P} = 0$, obtained with the $4d$ flat regulator. Results for several different initial values of $\bar{M}_{q,\Lambda}$ are compared. The initial values of four-quark couplings are fixed with $\bar{\lambda}_{\pi,\Lambda} = \bar{\lambda}_{\sigma,\Lambda} = 22.55$ and $\bar{\lambda}_{\alpha,\Lambda} = 0$ ($\alpha \notin \{\sigma, \pi\}$). Data points denote the results calculated in the flow equation in the Euclidean ($P^2 > 0$) region. The dashed and solid lines stand for results of analytic continuation from $P^2 > 0$ to $P^2 < 0$ based on the fit of the Padé[2,2] and Padé[20,20] approximants, respectively.

the analytic continuation. Hence we use the diagonal Padé approximants as in the 3d case. As there we have a rapid convergence, see Table 3: Already the Padé [2,2] approximant (dashed lines in Figure 15) agrees very well with the largest order [20,20] (straight lines in Figure 15) considered.

Table 3: Pion pole mass with different current quark masses, which are obtained from analytic continuation from the Euclidean to Minkowski regimes based on Padé approximants of different orders. Results in the Euclidean regions are calculated with the $4d$ flat regulator, where the initial values of four-quark couplings are fixed with $\bar{\lambda}_{\pi,k=\Lambda} = \bar{\lambda}_{\sigma,k=\Lambda} = 22.55$ and $\bar{\lambda}_{\alpha,k=\Lambda} = 0$ ($\alpha \notin \{\sigma, \pi\}$), see also Figure 15.

| $M_{q,k=\Lambda}[\Lambda]$ | $10^{-4}$ | $10^{-3}$ | $5 \times 10^{-3}$ | $2 \times 10^{-2}$ |
|---|---|---|---|---|
| Padé [1,1] | 0.05942 | 0.1064 | 0.1926 | 0.3225 |
| Padé [2,2] | 0.06313 | 0.1162 | 0.1971 | 0.3300 |
| Padé [10,10] | 0.06234 | 0.1219 | 0.2073 | 0.3442 |
| Padé [20,20] | 0.06361 | 0.1166 | 0.1966 | 0.3291 |

# 5  Conclusions

In this work we have investigated the quark mass production and the formation of bound states in the RG flows. This is done in a low energy effective theory of pure fermionic degrees of freedom within the fRG approach. The four-quark interaction is comprised of a Fierz-complete basis with ten channels, and thus there is no ambiguity arising from projecting the flows of four-quark vertices onto different channels. In the fRG approach flow equations of the quark propagator and the four-quark vertex play a similar role as the quark gap equation and Bethe-Salpeter equation for the four-quark scattering kernel in the formalism of DSE-BSE. Important relations resulting from the chiral symmetry and its dynamical breaking in low energy QCD, such as Gell-Mann–Oakes–Renner relation and Goldstone theorem, are guaranteed by the self-consistency between the flows of the two-point and four-point quark correlation functions.

The flow diagram of the quark mass and the four-quark couplings has been analysed in detail. In a reduction to the dominant four-quark coupling in of the scalar-pseudoscalar channel, the two-dimensional flow diagram is divided into a Gaußian regime and a regime with dynamical chiral symmetry breaking. In the Gaußian regime the $\beta$-function of coupling is positive for all cutoff scales, and the flow takes place in the attraction regime of the Gaußian fixed point. In the regime with dynamical chiral symmetry breaking the flow is initiated with a negative $\beta$-function for the coupling. This results in a significant dynamical enhancement of both the coupling and the quark mass during the flow.

We also have analysied the dominance of the $\sigma$ and $\pi$ channels over other channels in the Fierz-complete basis: we find that the differences between the results of the fully self-consistent calculation with ten Fierz-complete four-quark channels and the two-channel computation with $\lambda_{\sigma,k} \neq \lambda_{\pi,k}$ vary between 2-5% for all couplings and the quark mass, and hence are very small. This entails, that the feedback effects of non-dominant four-quark couplings, i.e., $\lambda_{\alpha,k}$ ($\alpha \notin \{\sigma, \pi\}$) are very small.

Finally, we have computed on-shell properties of pions within a Fierz complete computation with momentum-independent coupling except the resonant pseudoscalar channel. In the latter we have used a $t$-channel approximation. Bound states emerge naturally as poles of their respective four-quark scatterings when the momenta are in close vicinity to on-shell momenta. The pole mass of the pion is determined from both direct calculations in the Minkowski regime of momenta and the analytic continuation with based on results in the Euclidean regime. This allows us to check explicitly that Padé approximants work very well for determining the location of the (first) pole in the complex plain.

The approximation used in the present work is self-consistent, similar to the self-consistency of BSE-DSE bound state computations required for capturing the Goldstone theorem and hence the chiral limit. We have shown that the pion is indeed massless in the chiral limit. We emphasise that this self-consistency is not required in the fRG approach and the Goldstone theorem can naturally be built-in with dynamical hadronisation, independent of the approximations used.

The translation of this property into the present four-quark language as well as further conceptual and numerical improvements is subject of the follow-up works [40, 41]. Specifically, this series of works aims at QCD applications, and we hope to report the results in the near future.

# Acknowledgments

We thank Jens Braun, Gernot Eichmann, Fabian Rennecke, Nicolas Wink for discussions. This work is done within the fQCD collaboration,[1] and is supported by the National Natural Science Foundation of China under Grant No. 12175030, and EMMI. This work is funded by the Deutsche Forschungsgemeinschaft (DFG, German Research Foundation) under Germany's Excellence Strategy EXC 2181/1 - 390900948 (the Heidelberg STRUCTURES Excellence Cluster), the Collaborative Research Centre SFB 1225 (ISOQUANT).

# A    Fierz complete basis: two flavours

In the following, we list the four-quark interactions of ten channels. According to the varying properties under global transformations of the Dirac fields $SU_V(N_f)$, $U_V(1)$, $SU_A(N_f)$, and $U_A(1)$, the four-quark interaction can be classified into four different sets. The first set is comprised of four channels, to wit,

$$
\begin{aligned}
\mathcal{T}_{ijlm}^{(V-A)}\bar{q}_i q_l \bar{q}_j q_m &= (\bar{q}\gamma_\mu T^0 q)^2 - (\bar{q} i\gamma_\mu\gamma_5 T^0 q)^2\,,\\
\mathcal{T}_{ijlm}^{(V+A)}\bar{q}_i q_l \bar{q}_j q_m &= (\bar{q}\gamma_\mu T^0 q)^2 + (\bar{q} i\gamma_\mu\gamma_5 T^0 q)^2\,,\\
\mathcal{T}_{ijlm}^{(S-P)_+}\bar{q}_i q_l \bar{q}_j q_m &= (\bar{q}\,T^0 q)^2 - (\bar{q}\,\gamma_5 T^0 q)^2 + (\bar{q}\,T^a q)^2 - (\bar{q}\,\gamma_5 T^a q)^2\,,\\
\mathcal{T}_{ijlm}^{(V-A)^{\mathrm{adj}}}\bar{q}_i q_l \bar{q}_j q_m &= (\bar{q}\gamma_\mu T^0 t^a q)^2 - (\bar{q} i\gamma_\mu\gamma_5 T^0 t^a q)^2\,,
\end{aligned}
\tag{A.1a}
$$

which are invariant with all the aforementioned transformations. Here, $T^a$ and $t^a$ denote the generators of the flavor $SU(N_f)$ group and the color $SU(N_c)$ group, respectively. Here a summation for the index of generators is assumed. Moreover, there is the generator of $U(1)$ in the flavor space $T^0 = 1/\sqrt{2N_f}\,\mathbb{1}_{N_f\times N_f}$. The four-quark interactions in the second set read

$$
\begin{aligned}
\mathcal{T}_{ijlm}^{(S+P)_-}\bar{q}_i q_l \bar{q}_j q_m &= (\bar{q}\,T^0 q)^2 + (\bar{q}\,\gamma_5 T^0 q)^2 - (\bar{q}\,T^a q)^2 - (\bar{q}\,\gamma_5 T^a q)^2\,,\\
\mathcal{T}_{ijlm}^{(S+P)_-^{\mathrm{adj}}}\bar{q}_i q_l \bar{q}_j q_m &= (\bar{q}\,T^0 t^a q)^2 + (\bar{q}\,\gamma_5 T^0 t^a q)^2 - (\bar{q}\,T^a t^b q)^2 - (\bar{q}\,\gamma_5 T^a t^b q)^2\,,
\end{aligned}
\tag{A.1b}
$$

which preserve the symmetry of $SU_V(N_f)\otimes U_V(1)\otimes SU_A(N_f)$ while break $U_A(1)$. The third set includes another two channels as follows

$$
\begin{aligned}
\mathcal{T}_{ijlm}^{(S-P)_-}\bar{q}_i q_l \bar{q}_j q_m &= (\bar{q}\,T^0 q)^2 - (\bar{q}\,\gamma_5 T^0 q)^2 - (\bar{q}\,T^a q)^2 + (\bar{q}\,\gamma_5 T^a q)^2\,,\\
\mathcal{T}_{ijlm}^{(S-P)_-^{\mathrm{adj}}}\bar{q}_i q_l \bar{q}_j q_m &= (\bar{q}\,T^0 t^a q)^2 - (\bar{q}\,\gamma_5 T^0 t^a q)^2 - (\bar{q}\,T^a t^b q)^2 + (\bar{q}\,\gamma_5 T^a t^b q)^2\,.
\end{aligned}
\tag{A.1c}
$$

They are symmetric under the transformations of $SU_V(N_f)\otimes U_V(1)\otimes U_A(1)$ while break $SU_A(N_f)$. The fourth set consists of the last two channels, which read

$$
\begin{aligned}
\mathcal{T}_{ijlm}^{(S+P)_+}\bar{q}_i q_l \bar{q}_j q_m &= (\bar{q}\,T^0 q)^2 + (\bar{q}\,\gamma_5 T^0 q)^2 + (\bar{q}\,T^a q)^2 + (\bar{q}\,\gamma_5 T^a q)^2\,,\\
\mathcal{T}_{ijlm}^{(S+P)_+^{\mathrm{adj}}}\bar{q}_i q_l \bar{q}_j q_m &= (\bar{q}\,T^0 t^a q)^2 + (\bar{q}\,\gamma_5 T^0 t^a q)^2 + (\bar{q}\,T^a t^b q)^2 + (\bar{q}\,\gamma_5 T^a t^b q)^2\,.
\end{aligned}
\tag{A.1d}
$$

They only preserve $SU_V(N_f)\otimes U_V(1)$ while break both $SU_A(N_f)$ and $U_A(1)$.

Finally, we note that all the tensors defined in this Appendix carry the Grassmann symmetries of the product of quarks and anti-quarks attached,

$$
\mathcal{T}_{ijlm}^{(\alpha)} = -\mathcal{T}_{jilm}^{(\alpha)} = -\mathcal{T}_{ijml}^{(\alpha)} = \mathcal{T}_{jiml}^{(\alpha)}\,.
\tag{A.2}
$$

[1]The fQCD collaboration is comprised of J. Braun, Y.-r. Chen, W.-j. Fu, F. Gao, A. Geissel, J. Horak, C. Huang, F. Ihssen, J. M. Pawlowski, F. Rennecke, F. Sattler, B. Schallmo, Y.-y. Tan, S. Topfel, R. Wen, J. Wessely, N. Wink and S. Yin.

# B Flow equations of the quark two-point and four-point correlation functions

The flow equation for the quark wave function $Z_q(p)$ in (9) is readily obtained by projecting the flow of the quark two-point function in the first row in Figure 4 onto the vector channel, that reads

$$
\begin{aligned}
\partial_t Z_q(p) = \int \frac{d^4q}{(2\pi)^4} \big(Z_q^R(q)\tilde{\partial}_t \bar{G}_q(q) + \bar{G}_q(q)\tilde{\partial}_t Z_q^R(q)\big)\frac{p\cdot q}{p^2} \\
\times \Big[\frac{3}{2}\lambda_\pi(-p,-q,p,q) + \frac{1}{2}\lambda_\sigma(-p,-q,p,q) - \frac{3}{2}\lambda_a(-p,-q,p,q) \\
- \frac{1}{2}\lambda_\eta(-p,-q,p,q) - \frac{8}{3}\lambda_{(S-P)_-^{\mathrm{adj}}}(-p,-q,p,q) - 12\lambda_{(V+A)}(-p,-q,p,q) \\
- 14\lambda_{(V-A)}(-p,-q,p,q) - \frac{8}{3}\lambda_{(V-A)^{\mathrm{adj}}}(-p,-q,p,q)\Big],
\end{aligned}
\tag{B.1a}
$$

where the explicit expressions of $\bar{G}_q(q)$, $Z_q^R(q)$, $\tilde{\partial}_t \bar{G}_q(q)$, $\tilde{\partial}_t Z_q^R(q)$ are given in Eqs. (C.2) through (C.5). Similarly, projecting the same flow onto the scalar channel, one arrives at the flow equation of the quark mass function, as follows

$$
\begin{aligned}
\partial_t M_q(p) = -\frac{\partial_t Z_q(p)}{Z_q(p)}M_q(p) + \int \frac{d^4q}{(2\pi)^4}\big(\tilde{\partial}_t \bar{G}_q(q)\big)M_q(q) \\
\times \Big[\frac{3}{2}\lambda_\pi(-p,-q,p,q) + \frac{23}{2}\lambda_\sigma(-p,-q,p,q) - \frac{3}{2}\lambda_a(-p,-q,p,q) \\
+ \frac{1}{2}\lambda_\eta(-p,-q,p,q) + \frac{8}{3}\lambda_{(S+P)_-^{\mathrm{adj}}}(-p,-q,p,q) \\
- \frac{16}{3}\lambda_{(S+P)_+^{\mathrm{adj}}}(-p,-q,p,q) - 4\lambda_{(V+A)}(-p,-q,p,q)\Big].
\end{aligned}
\tag{B.1b}
$$

The flow equations of the four-quark dressings $\lambda_\alpha$'s read

$$
\begin{aligned}
& \partial_t \lambda_\alpha(p_1,p_2,p_3,p_4) \\
& = \sum_{\alpha',\alpha''} \int \frac{d^4q}{(2\pi)^4}\Big[\lambda_{\alpha'}(p_1,-p_1-q-p_3,p_3,q)\lambda_{\alpha''}(p_2,-q,p_4,q+p_3+p_1)\mathcal{F}_{\alpha'\alpha'',\alpha}^t \\
& \qquad + \lambda_{\alpha'}(p_2,-p_2-q-p_3,p_3,q)\lambda_{\alpha''}(p_1,-q,p_4,q+p_3+p_2)\mathcal{F}_{\alpha'\alpha'',\alpha}^u \\
& \qquad + \lambda_{\alpha'}(p_1,p_2,q,-q-p_1-p_2)\lambda_{\alpha''}(-q,q+p_1+p_2,p_3,p_4)\mathcal{F}_{\alpha'\alpha'',\alpha}^s\Big],
\end{aligned}
\tag{B.1c}
$$

where the three terms in the square bracket on the r.h.s. of (B.1c) above correspond to the $t$-, $u$-, and $s$-channels of loop diagrams in the flow equation of the four-point vertex as shown in the second line of Figure 4, respectively. The coefficients $\mathcal{F}_{\alpha'\alpha'',\alpha}^t$, $\mathcal{F}_{\alpha'\alpha'',\alpha}^u$, and $\mathcal{F}_{\alpha'\alpha'',\alpha}^s$ are momentum-dependent functions of quark propagators and regulators. In Appendix C relations among these coefficients are discussed, and explicit expressions of some selective coefficients are presented.

# C Coefficients of the Fierz complete flow

In this appendix we present explicit expressions of some coefficients $\mathcal{F}_{\alpha'\alpha'',\alpha}^t$, $\mathcal{F}_{\alpha'\alpha'',\alpha}^u$, and $\mathcal{F}_{\alpha'\alpha'',\alpha}^s$ in (B.1c), which are restricted to $\alpha,\alpha',\alpha'' \in \{\sigma,\pi\}$ for illustrative purpose. We begin

with

$$\mathcal{F}^t_{\sigma\sigma,\sigma} = -\frac{1}{N_f}\big(\tilde{\partial}_t \bar{G}_q(q)\big)\bar{G}_q(q+p_1+p_3)\Big[21 M_q(q)M_q(q+p_1+p_3)Z_q(q)Z_q(q+p_1+p_3)$$

$$-22 q\cdot(q+p_1+p_3)Z_q^R(q)Z_q^R(q+p_1+p_3)\Big]$$

$$= -\frac{1}{N_f}\bar{G}_q(q)\bar{G}_q(q+p_1+p_3)(-22)q\cdot(q+p_1+p_3)\big(\tilde{\partial}_t Z_q^R(q)\big)Z_q^R(q+p_1+p_3),\quad\text{(C.1)}$$

where one used the notations as follows

$$Z_q^R(q) = Z_q(q) + R_F(q)\,,\tag{C.2}$$

with $R_F(q) = Z_q(q)r_q(q^2/k^2)$, and

$$\bar{G}_q(q) = \frac{1}{\big(Z_q^R(q)\big)^2 q^2 + Z_q^2(q)M_q^2(q)}\,,\tag{C.3}$$

as well as

$$\tilde{\partial}_t \bar{G}_q(q) = -2\big(\bar{G}_q(q)\big)^2 Z_q^R(q)q^2 \partial_t R_F(q)\,,\tag{C.4}$$

$$\tilde{\partial}_t Z_q^R(q) = \partial_t R_F(q)\,.\tag{C.5}$$

To proceed, one arrives at

$$\mathcal{F}^t_{\pi\pi,\sigma} = \frac{N_f}{12(N_f+2)}\big(\tilde{\partial}_t \bar{G}_q(q)\big)\bar{G}_q(q+p_1+p_3)$$

$$\times\Big[9(N_f+2)M_q(q)M_q(q+p_1+p_3)Z_q(q)Z_q(q+p_1+p_3)$$

$$+2q\cdot(q+p_1+p_3)Z_q^R(q)Z_q^R(q+p_1+p_3)\Big]\tag{C.6}$$

$$+\frac{N_f}{12(N_f+2)}\bar{G}_q(q)\bar{G}_q(q+p_1+p_3)2q\cdot(q+p_1+p_3)\big(\tilde{\partial}_t Z_q^R(q)\big)Z_q^R(q+p_1+p_3)\,,$$

and

$$\mathcal{F}^t_{\sigma\pi,\sigma} = \frac{1}{6(N_f+2)}\big(\tilde{\partial}_t \bar{G}_q(q)\big)\bar{G}_q(q+p_1+p_3)$$

$$\times\Big[-9(N_f+2)M_q(q)M_q(q+p_1+p_3)Z_q(q)Z_q(q+p_1+p_3)$$

$$+(9N_f+17)q\cdot(q+p_1+p_3)Z_q^R(q)Z_q^R(q+p_1+p_3)\Big]$$

$$+\frac{1}{6(N_f+2)}\bar{G}_q(q)\bar{G}_q(q+p_1+p_3)(9N_f+17)$$

$$\times q\cdot(q+p_1+p_3)\big(\tilde{\partial}_t Z_q^R(q)\big)Z_q^R(q+p_1+p_3)\,.\tag{C.7}$$

Moreover, it is found that

$$\mathcal{F}^t_{\pi\sigma,\sigma} = \mathcal{F}^t_{\sigma\pi,\sigma}\,,\tag{C.8}$$

which is demanded by the symmetry properties of $\lambda_\alpha$'s in (15). Insofar as the pion basis is concerned, one arrives at

$$\mathcal{F}^t_{\sigma\sigma,\pi} = 0\,,\tag{C.9}$$

and

$$
\mathcal{F}^t_{\pi\pi,\pi} = \frac{1}{6(N_f+2)}\big(\tilde{\partial}_t \bar{G}_q(q)\big)\bar{G}_q(q+p_1+p_3)
$$
$$
\times \Big[ 78(N_f+2)M_q(q)M_q(q+p_1+p_3)Z_q(q)Z_q(q+p_1+p_3)
$$
$$
+ (77N_f+156)q\cdot(q+p_1+p_3)Z_q^R(q)Z_q^R(q+p_1+p_3)\Big]
$$
$$
+ \frac{1}{6(N_f+2)}\bar{G}_q(q)\bar{G}_q(q+p_1+p_3)(77N_f+156)
$$
$$
\times q\cdot(q+p_1+p_3)\big(\tilde{\partial}_t Z_q^R(q)\big)Z_q^R(q+p_1+p_3)\,, \tag{C.10}
$$

as well as

$$
\mathcal{F}^t_{\sigma\pi,\pi} = \mathcal{F}^t_{\pi\sigma,\pi} = \frac{1}{6N_f(N_f+2)}\big(\tilde{\partial}_t \bar{G}_q(q)\big)\bar{G}_q(q+p_1+p_3)
$$
$$
\times \Big[ 12(N_f+2)M_q(q)M_q(q+p_1+p_3)Z_q(q)Z_q(q+p_1+p_3)
$$
$$
+ (7N_f+12)q\cdot(q+p_1+p_3)Z_q^R(q)Z_q^R(q+p_1+p_3)\Big]
$$
$$
+ \frac{1}{6N_f(N_f+2)}\bar{G}_q(q)\bar{G}_q(q+p_1+p_3)(7N_f+12)
$$
$$
\times q\cdot(q+p_1+p_3)\big(\tilde{\partial}_t Z_q^R(q)\big)Z_q^R(q+p_1+p_3)\,. \tag{C.11}
$$

Coefficients of the *u*-channel are related to those of the *t*-channel through the symmetry relation of $\lambda_\alpha$'s in (15), under the interchange of the momenta of two quarks or two antiquarks. As a consequence, one is led to

$$
\mathcal{F}^u_{\alpha'\alpha'',\alpha} = \mathcal{F}^t_{\alpha'\alpha'',\alpha}\Big|_{p_1 \to p_2}\,. \tag{C.12}
$$

Finally, we show some coefficients for the *s*-channel:

$$
\mathcal{F}^s_{\sigma\sigma,\sigma} = \frac{2}{N_f}\big(\tilde{\partial}_t \bar{G}_q(q)\big)\bar{G}_q(-q-p_1-p_2)M_q(q)M_q(-q-p_1-p_2)Z_q(q)Z_q(-q-p_1-p_2)\,, \tag{C.13}
$$

and

$$
\mathcal{F}^s_{\pi\pi,\sigma} = \frac{N_f}{6(N_f+2)}\big(\tilde{\partial}_t \bar{G}_q(q)\big)\bar{G}_q(-q-p_1-p_2)
$$
$$
\times \Big[ 9(N_f+2)M_q(q)M_q(-q-p_1-p_2)Z_q(q)Z_q(-q-p_1-p_2)
$$
$$
- 2q\cdot(-q-p_1-p_2)Z_q^R(q)Z_q^R(-q-p_1-p_2)\Big]
$$
$$
+ \frac{N_f}{6(N_f+2)}\bar{G}_q(q)\bar{G}_q(-q-p_1-p_2)(-2)
$$
$$
\times q\cdot(-q-p_1-p_2)\big(\tilde{\partial}_t Z_q^R(q)\big)Z_q^R(-q-p_1-p_2)\,, \tag{C.14}
$$

moreover,

$$
\mathcal{F}^s_{\sigma\pi,\sigma} = \mathcal{F}^s_{\pi\sigma,\sigma} = \frac{1}{3(N_f+2)}\big(\tilde{\partial}_t \bar{G}_q(q)\big)\bar{G}_q(-q-p_1-p_2)q \tag{C.15}
$$
$$
\cdot(-q-p_1-p_2)Z_q^R(q)Z_q^R(-q-p_1-p_2)
$$
$$
+ \frac{1}{3(N_f+2)}\bar{G}_q(q)\bar{G}_q(-q-p_1-p_2)q \tag{C.16}
$$
$$
\cdot(-q-p_1-p_2)\big(\tilde{\partial}_t Z_q^R(q)\big)Z_q^R(-q-p_1-p_2)\,. \tag{C.17}
$$

In the same way, for the pion basis one arrives at

$$\mathcal{F}^s_{\sigma\sigma,\pi} = 0 \,, \tag{C.18}$$

and

$$\mathcal{F}^s_{\pi\pi,\pi} = \frac{N_f}{3(N_f+2)}\big(\tilde{\partial}_t \bar{G}_q(q)\big)\bar{G}_q(-q-p_1-p_2)q\cdot(-q-p_1-p_2)Z^R_q(q)Z^R_q(-q-p_1-p_2)$$
$$+\frac{N_f}{3(N_f+2)}\bar{G}_q(q)\bar{G}_q(-q-p_1-p_2)q\cdot(-q-p_1-p_2)\big(\tilde{\partial}_t Z^R_q(q)\big)Z^R_q(-q-p_1-p_2)\,, \tag{C.19}$$

also

$$\mathcal{F}^s_{\sigma\pi,\pi} = \mathcal{F}^s_{\pi\sigma,\pi} = \frac{1}{3N_f(N_f+2)}\big(\tilde{\partial}_t \bar{G}_q(q)\big)\bar{G}_q(-q-p_1-p_2)$$
$$\times\Big[6(N_f+2)M_q(q)M_q(-q-p_1-p_2)Z_q(q)Z_q(-q-p_1-p_2)$$
$$-N_f q\cdot(-q-p_1-p_2)Z^R_q(q)Z^R_q(-q-p_1-p_2)\Big]$$
$$+\frac{1}{3N_f(N_f+2)}\bar{G}_q(q)\bar{G}_q(-q-p_1-p_2)(-N_f)q$$
$$\cdot(-q-p_1-p_2)\big(\tilde{\partial}_t Z^R_q(q)\big)Z^R_q(-q-p_1-p_2)\,. \tag{C.20}$$

## D  Dominance of the scalar-pseudoscalar four-quark channel

In this appendix we demonstrate the dominance of the $\sigma$ and $\pi$ channels in the Fierz-complete four-quark couplings of ten channels, as discussed in Section 3.2. Two computations are compared in detail: One is the fully self-consistent calculation with ten Fierz-complete four-quark channels, and the other one is the two-channel computation with only the $\sigma$ and $\pi$ couplings. Note that in the two-channel calculation, the $\sigma$ and $\pi$ couplings are distinguished, viz., $\lambda_{\sigma,k} \neq \lambda_{\pi,k}$ for a generic cutoff scale $k$, which is different from the case of the one-channel truncation as discussed in Section 3.1. Furthermore, we also investigate the feedback effects of four-quark couplings of other channels, i.e., $\lambda_{\alpha,k}$ ($\alpha \notin \{\sigma,\pi\}$) on the running of flows in a truncation, which is called as the quenched calculation in this work. In the quenched calculation, the four-quark dressings $\lambda_{\sigma,k}$ and $\lambda_{\pi,k}$, and the mass function obtained in the two-channel computation as well as $\lambda_{\alpha,k} = 0$ ($\alpha \notin \{\sigma,\pi\}$) are input into the r.h.s. of the four-quark flow equations for other channels. In another word, the four-quark couplings of other channels are decoupled from the coupled flow equations, and their nonvanishing values are completely resulting from their respective flow diagrams in the scalar-pseudoscalar channels.

The quark mass and the four-quark couplings of the $\sigma$ and $\pi$ channels are compared between these two computations in Figure 16 and Figure 17, respectively. It is observed that the difference between the full results and the two-channel ones is less than 2% for the constituent quark mass, less than 3% for $\lambda_\sigma$ and less than 5% for $\lambda_\pi$. Moreover, it is found that the feedback effects of non-dominant four-quark couplings, i.e., $\lambda_{\alpha,k}$ ($\alpha \notin \{\sigma,\pi\}$) are negligible, as shown in Figure 18, where the full and quenched results are compared.

## E  Coefficients of Equation (37)

In the $t$-channel approximation described in (36), the flow for $\lambda_\pi$ is given by (37) with the coefficients $\mathcal{A}, \mathcal{B}, \mathcal{C}$.

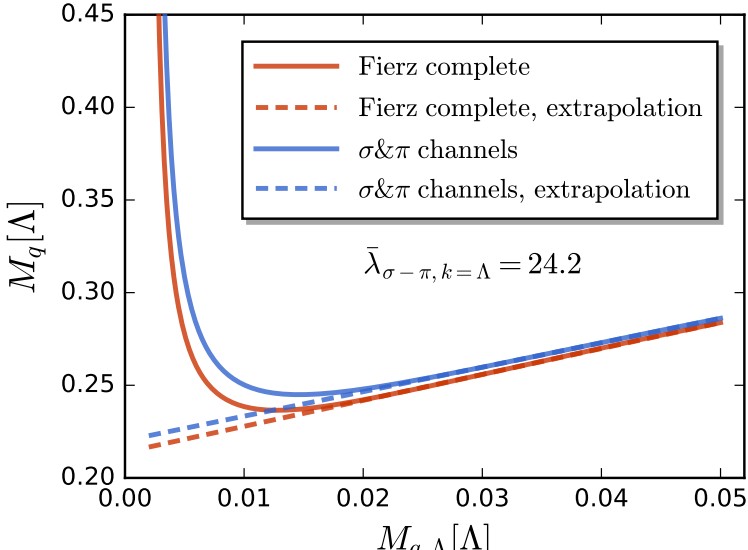

Figure 16: Constituent quark mass $M_{q,k\to 0}$ as a function of the current quark mass $M_{q,\Lambda}$ obtained from the computation of the Fierz-complete basis (red solid) and that of the two channels with $\sigma$ and $\pi$, cf. (16). The same initial conditions of the four-quark couplings $\bar{\lambda}_{\pi,\Lambda} = \bar{\lambda}_{\sigma,\Lambda} = 24.2$ and $\bar{\lambda}_{\alpha,\Lambda} = 0$ ($\alpha \notin \{\sigma, \pi\}$) are used for the two computations. The respective linear extrapolations towards the chiral limit are depicted as dashed lines.

The coefficient $\mathcal{A}$ is readily obtained from the r.h.s. of (B.1c),

$$\mathcal{A}(P) = \int \frac{d^4q}{(2\pi)^4} \sum_{\substack{\alpha \neq \pi \\ \alpha' \neq \pi}} \lambda_{\alpha,k}\, \lambda_{\alpha',k} \left[ \mathcal{F}^t_{\alpha\alpha',\pi}(P) + \mathcal{F}^u_{\alpha\alpha',\pi} + \mathcal{F}^s_{\alpha\alpha',\pi} \right], \tag{E.1}$$

with momentum independent four-quark couplings $\lambda_{\alpha \neq \pi}$. The momentum dependence of $\mathcal{A}(P)$ comes solely from the first diagram on the r.h.s. in the second line in Figure 4 as the loops $\mathcal{F}^{s,u}_{\alpha\alpha',\pi}$ only depend on $s, u = 0$ in the present momentum configuration.

The coefficient $\mathcal{B}$ is given by

$$\mathcal{B}(P) = 2 \int \frac{d^4q}{(2\pi)^4} \sum_{\substack{\alpha' \neq \pi \\ \alpha'' = \pi}} \lambda_{\alpha',k} \left[ \mathcal{F}^t_{\alpha'\alpha'',\pi}(P) + \mathcal{F}^u_{\alpha'\alpha'',\pi} + \mathcal{F}^s_{\alpha'\alpha'',\pi} \right], \tag{E.2}$$

where the factor 2 on the r.h.s. comes from the interchange of the indices $\alpha'$ and $\alpha''$, and as for $\mathcal{A}$ its momentum dependence is due to the $t$-channel diagram in Figure 4. Finally, the most relevant coefficient $\mathcal{C}$, see (C.10), is given by

$$\mathcal{C}(P) = \int \frac{d^4q}{(2\pi)^4} \mathcal{F}^t_{\pi\pi,\pi}(P). \tag{E.3}$$

It only comprises the contributions from the $t$-channel diagram in Figure 4.

# F  Sub-leading four-quark vertices

As we have discussed in Section 3, when the momentum dependence of four-quark dressings $\lambda_\alpha$ is neglected, there is a Fierz complete basis of four-quark interactions, which includes

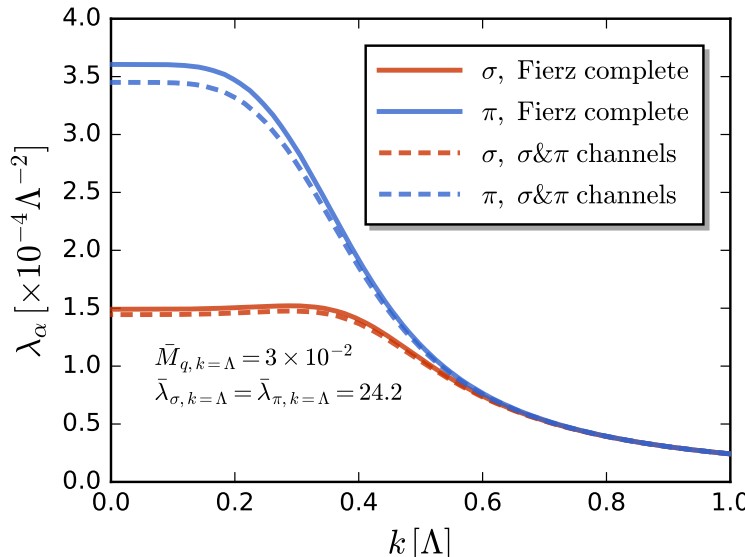

Figure 17: Four-quark couplings of the $\sigma$ and $\pi$ channels as functions of the RG scale $k$. Results obtained from the computation of the Fierz-complete basis are compared with those from the two-channel computation. The same initial conditions $\bar{\lambda}_{\pi,\Lambda} = \bar{\lambda}_{\sigma,\Lambda} = 24.2$, $\bar{\lambda}_{\alpha,\Lambda} = 0$ ($\alpha \notin \{\sigma, \pi\}$), and $\bar{M}_{q,\Lambda} = 3 \times 10^{-2}$ are used for the two computations.

ten tensors $\mathcal{T}^{(\alpha)}_{ijlm}$ with $\alpha = 1, ..., 10$ for the two flavor case, presented in Appendix A. The Grassmann nature of quark fields and the momentum-independence of $\lambda_\alpha$ lead us to the anti-symmetric properties of the ten tensors under the interchange of indices connected two quarks or antiquarks, to wit,

$$\mathcal{T}^{(\alpha)}_{ijlm} = -\mathcal{T}^{(\alpha)}_{jilm} = -\mathcal{T}^{(\alpha)}_{ijml} = \mathcal{T}^{(\alpha)}_{jiml}, \tag{F.1}$$

with $\alpha = 1, ..., 10$. Evidently, when the momentum dependence of four-quark dressings is taken into account, one can easily extend the ten tensors to another ten ones, here denoted by $\mathcal{T}^{(\bar{\alpha})}_{ijlm}$ with $\bar{\alpha} = 11, ..., 20$, which are symmetric under the interchange of two quarks or antiquarks, i.e.,

$$\mathcal{T}^{(\bar{\alpha})}_{ijlm} = \mathcal{T}^{(\bar{\alpha})}_{jilm} = \mathcal{T}^{(\bar{\alpha})}_{ijml} = \mathcal{T}^{(\bar{\alpha})}_{jiml}, \tag{F.2}$$

with $\bar{\alpha} = 11, ..., 20$. Since the combination $\lambda^{(\bar{\alpha})}(\boldsymbol{p}) \mathcal{T}^{(\bar{\alpha})}_{ijlm}$ has to be anti-symmetric under the interchange of two quarks or antiquarks, one arrives at

$$\lambda_{\bar{\alpha}}(p_1, p_2, p_3, p_4) = -\lambda_{\bar{\alpha}}(p_2, p_1, p_3, p_4) = -\lambda_{\bar{\alpha}}(p_1, p_2, p_4, p_3) = \lambda_{\bar{\alpha}}(p_2, p_1, p_4, p_3), \tag{F.3}$$

with $\bar{\alpha} = 11, ..., 20$, which is a natural extension of the symmetry properties for the first ten four-quark dressings in (15). Note that from (F.3) it is readily found that $\lambda_{\bar{\alpha}}$ are vanishing when their momentum dependence is neglected, which thus can be regarded as sub-leading four-quark vertices in terms of the expansion of external momenta.

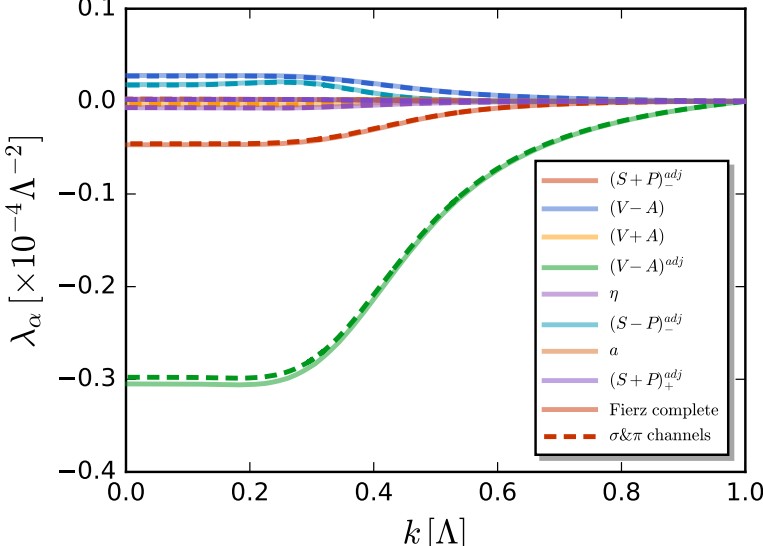

Figure 18: Non-dominant four-quark couplings of the Fierz-complete channels, i.e., $\lambda_{\alpha,k}$ ($\alpha \notin \{\sigma, \pi\}$), as functions of the RG scale $k$. Results obtained from the self-consistent Fierz-complete computation (solid lines) are compared the quenched results. In the quenched calculation, the four-quark dressings $\lambda_{\sigma,k}$ and $\lambda_{\pi,k}$ obtained in the two-channel computation as well as $\lambda_{\alpha,k} = 0$ ($\alpha \notin \{\sigma, \pi\}$) are input into the r.h.s. of the four-quark flow equations for other channels. The same initial conditions as Figure 17 are employed.

## G  Regulator dependence of the bound state results

The results in the present work have been obtained with the flat or Litim regulator, whose shape function is given by

$$r_q^{\text{flat}}(x) = \left(\frac{1}{\sqrt{x}} - 1\right)\Theta(1-x), \tag{G.1}$$

with the Heaviside step function $\Theta(x)$. This regulator is a convenient choice within simple, momentum-independent approximations. Moreover, it is the optimal regulator for momentum-independent truncations, see [25, 49, 50]. In the presence of wave functions ($Z_k$ or $Z_k(p)$) it is not optimal any more, [25]. Still it is a convenient choice as it leads to analytic flows, and we shall also used it for our analysis of chiral symmetry breaking with scale-dependent wave function, mass, and four point vertices.

For momentum-dependent approximations including scale-dependent wave functions $Z_k$ it looses both properties: it neither leads to analytic flows nor is it optimal, see [25] for the systematic construction of optimal regulators for higher orders of the derivative expansion or momentum-dependent approximations. Moreover, the non-analyticity of the regulator even slows down standard integration routines.

For the sake of comparability with the results on chiral symmetry breaking we have still used the flat regulator for our bound state analysis. A self-consistency check of these results is done by varying the shape function $r_q$ of the regulator. We use a simple exponential regulator,

$$r_q^{\text{exp}}(x) = \frac{1}{x}e^{-x}, \qquad x = \frac{p^2}{k^2}, \quad \text{or} \quad x = \frac{\boldsymbol{p}^2}{k^2}. \tag{G.2}$$

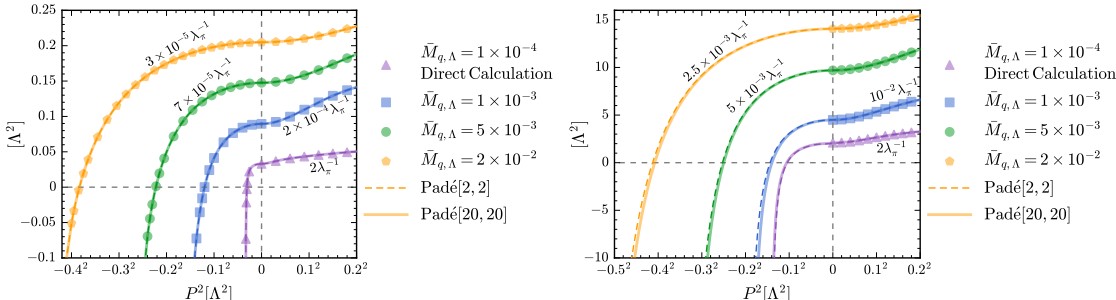

Figure 19: Left panel: Inverse four-quark coupling of the $\pi$ channel, $1/\lambda_{\pi,k=0}$, as a function of the Mandelstam variable $t = P^2 = P_0^2 + \boldsymbol{P}^2$ with $\boldsymbol{P} = 0$, which is the analogue of results in the left panel of Figure 14, but calculated with the $3d$ exponential regulator in (G.2). Here the initial four-quark couplings are chosen to be $\bar{\lambda}_{\pi,k=\Lambda} = \bar{\lambda}_{\sigma,k=\Lambda} = 6.11$ and $\bar{\lambda}_{\alpha,k=\Lambda} = 0$ ($\alpha \notin \{\sigma, \pi\}$). Right panel: Inverse four-quark coupling of the $\pi$ channel, $1/\lambda_{\pi,k=0}$, as a function of the Mandelstam variable $t = P^2 = P_0^2 + \boldsymbol{P}^2$ with $\boldsymbol{P} = 0$, which is the analogue of results in Figure 15, obtained with the $4d$ exponential regulator in (G.2). Here the initial four-quark couplings are chosen to be $\bar{\lambda}_{\pi,k=\Lambda} = \bar{\lambda}_{\sigma,k=\Lambda} = 9.98$ and $\bar{\lambda}_{\alpha,k=\Lambda} = 0$ ($\alpha \notin \{\sigma, \pi\}$).

Equation (G.2) and variants thereof are common choices for momentum-dependent approximations, for applications in QCD see e.g. [18, 20, 48, 58].

As indicated in (G.2), we have used both, the spatial momentum ($3d$) and $4d$ shape functions in the bound state computations for a comparison with the results obtained with the flat regulator (G.1). The use of the $3d$ shape function is relevant for its application to QCD at finite temperature and density, where such regulators carry the Silver blaze property [29, 59–63].

We employ the exponential regulator in (G.2) and redo the calculations of bound states in the left panel of Figure 14 and in Figure 15 for the $3d$ and $4d$ cases, respectively, and the relevant results are presented in Figure 19. One can see the exponential regulator only results in minor quantitative distinctions in comparison to the results obtained from the flat regulator, in both the $3d$ and $4d$ cases. Thus, one concludes that the bound state results obtained in the RG flows within the one-momentum-channel truncation in this work, show no obvious dependence or preference on what kinds of regulators are used.

Moreover, the Lorentz invariance is broken by the $3d$ regulators, while it is preserved by the $4d$ one. Therefore, it is possible to investigate how large the breaking effect of Lorentz invariance by the $3d$ regulators is, through a direct comparison between the $3d$ and $4d$ results. The relevant comparison is done in Figure 20, where the inverse four-quark coupling of the $\pi$ channel is depicted as a function of $P_0^2$ in the Euclidean regime with the $3d$ and $4d$ flat regulators. Similar results are also found for the exponential regulators. The comparison is done for several different initial values of the quark mass, and for each value of the quark mass, the initial four-quark couplings are adjusted a bit, such that both the $3d$ and $4d$ calculations produce the same results at $P_0 = 0$. One can see that the comparison in Figure 20 indicates that the breaking effect of Lorentz invariance resulting from the $3d$ regulators is mild.

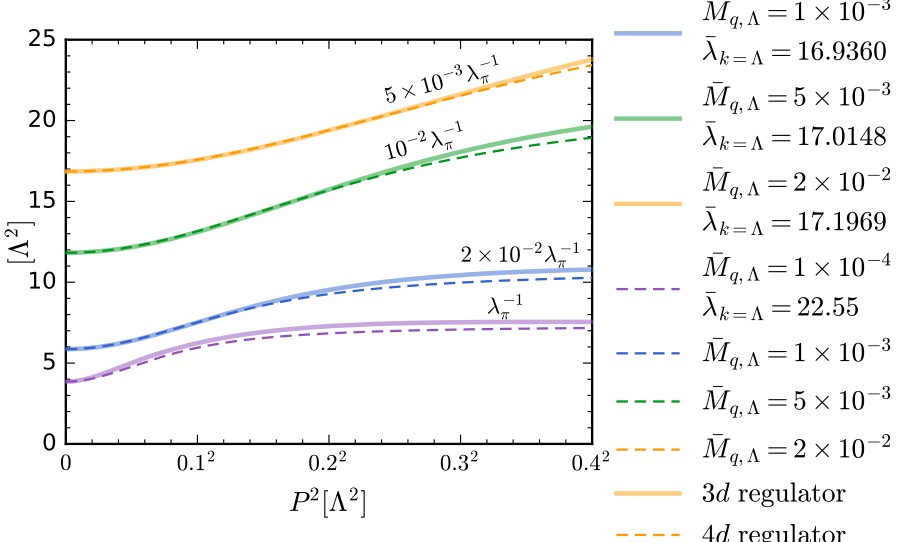

Figure 20: Comparison of the inverse four-quark coupling $1/\lambda_{\pi,k=0}$ between the results obtained with the $3d$ regulator (solid line) and those with the $4d$ one (dashed line), where the flat shape function has been used for both calculations. Several different initial values of $\bar{M}_{q,k=\Lambda}$ are adopted. The initial four-quark coupling $\bar{\lambda}_{k=\Lambda} = \bar{\lambda}_{\pi,k=\Lambda} = \bar{\lambda}_{\sigma,k=\Lambda}$ with $\bar{\lambda}_{\alpha,k=\Lambda} = 0$ ($\alpha \notin \{\sigma, \pi\}$) is adjusted a bit, such that both the $3d$ and $4d$ calculations arrive at the same $1/\lambda_{\pi,k=0}$ at $P_0 = 0$ for each value of the current quark mass.

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
