# Peer review of "Four-quark scatterings in QCD I"

_SciPost Physics, doi:SciPost Phys. 14, 069 (2023)_

## Round 1 · Referee Report · Anonymous (Referee 1) · 2022-10-27

Report

Dear editor and authors,

I have read the article, and it is both an interesting and worthwhile work. It is certainly publishable in its current form. However, the progress in this work is essentially incremental only, and mainly sets up a framework. But it does in its calculation only go somewhat beyond the results established using FRGs previously. Though it is, of course, nice to make a first foray into Minkowski space-time, the real complication, namely that gluons may very well be not surpressed off-shell, is not really addressed, especially what is infrared, once there are lightlike directions. Even when attempting, as is done, to use a 3d regulator only, the reason why it works is the very specific case and momentum configuration considered. At least to me it is not clear how this should be transcended from the available information. Thus, a publication in SciPost Core may be more adequate.

Moreover, the authors should consider optionally to streamline their paper. Right now, it has a lot of redundant information I have read in FRG papers several times over. Also, a lot of side remarks are so far from directly connected to the main part of the work, or have been reiterated so often in the literature, that they appear to serve only the purpose to add further references. I would really suggest to the authors to reconsider this style. But again, this is optional on my part, as the authors have the right to present their work in any way they find suitable. It is just the impression I get.
  • validity: high
  • significance: ok
  • originality: good
  • clarity: good
  • formatting: excellent
  • grammar: excellent

Author:  Chuang Huang  on 2022-12-02  [id 3098]

(in reply to Report 2 on 2022-10-27)

We would like to thank the referee for his comments on this manuscript. There is the response as follows.

We completely do not agree with the statement by the referee in Report 2, that in this work we just set up a framework and the progress is essentially incremental only. Yes, we have set up a framework, but it is not trivial. It has never been known before that the coupled system of the flow of the four-quark vertex and that of the quark two-point function can play the same role as the system of the Bethe-Salpeter equation and the quark gap equation. That is the reason why the referee in Report 1 says ''It is very interesting to me the conclusion that by computing the functional renormalisation group flows of the Fierz-complete four-quark interaction of up and down quarks with its t channel momentum dependence (in the isospin symmetric case, also including the flow of the quark two-point function) the system can be understood as the fRG analogues of the complete Bethe-Salpeter (BS) equations and quark gap equation.'' and ''I think this article will be a further step on the road to a better understanding of QCD.''

In fact, we found by accident that the quark mass production and the natural emergence of bound states can be well described by the language of the flows of four-quark and two-quark correlation functions. Therefore, in the first paper of this sequence we try to demonstrate clearly the underlying mechanism by employing the simplest truncation, that is, the momentum-independent quark mass and just one-channel momentum-dependent four-quark vertex. Even within this simplest truncation, one can see that the fRG analogues of BS and gap equations provide us with the necessary ingredients to study the physics related to the dynamical chiral symmetry breaking. Hence, the mechanism and feasibility of the framework are the central concerns in the first paper. We have no idea why the referee in Report 2 focuses on the stuffs of Minkowski space-time and 3d regulator. The reason why we use the 3d regulator in the work is that, the results of 3d regulator directly computed in the Minkowski space-time are in good agreement with those of analytic continuation from the Euclidean calculations as shown in the left panel of Fig.14, which shows that meson observables, e.g., the pole mass of pion, can be reliably extracted via analytic continuation based on results in the Euclidean regime. We certainly know that with more sophisticated truncations, such as the more momentum dependence for the four-quark and two-quark vertices in the second paper of this sequence, and encoding of the gluon dynamics in this framework in Paper III, one is not able to do the calculations directly in the Minkowski regime. But one can use the verified method of analytic continuation to extract observables in Minkowski regime.

Moreover, we have obtained some preliminary results in Paper II and Paper III. For example, in Paper II we employ a momentum-dependent quark mass and three-momentum $s$-, $t$-, $u$-channel dependent four-quark vertices, and obtain the poles masses of not only pion but also the scalar meson sigma, the pion decay constant, the Bethe-Salpeter amplitude of pion, etc. In Paper III, the flow of the four-quark vertex is included in QCD self-consistently, which receives contributions from not only the four-quark scattering but also the gluon exchanges. We hope we can report the relevant results in the near future.

Since this is the first paper of series, we would like to discuss more about the relation between the new framework with e.g., the dynamical hadronization. Therefore, we would like to keep it as it is.

---

## Round 1 · Referee Report · Anonymous (Referee 2) · 2022-11-16

Strengths

1- Impressive work on an very relevant topic: the study of dynamical chiral symmetry breaking and the emergence of mesonic bound states from the infrared dynamics of four-quark scatterings by using functional methods for low energy QCD. 2- The manuscript is clear and well written. 3- Very good technical appendices. 4- Results that are the basis for further studies in the future.

Report

Report 1 (11-2022)

The manuscript is the first of a sequence of papers where the Authors study the four-quark scatterings in QCD, being the second a third parts the study of full momentum dependences of the s, t, u channels as well as the discussion of the remnant momentum dependence, to be done in [40], and the embedding in QCD, being done in [41]. This is a massive task. The present work is a complement of the functional renormalisation group approach for bound state computations in QCD (where some of the Authors gave a very important contribution, namely in [5] where QCD phase structure at finite temperature and density for N_f=2+1 was studied). By considering two flavor QCD the Authors investigate dynamical chiral symmetry breaking and the emergence of mesonic bound states from the infrared dynamics of four-quark scatterings. It is very interesting to me the conclusion that by computing the functional renormalisation group flows of the Fierz-complete four-quark interaction of up and down quarks with its t channel momentum dependence (in the isospin symmetric case, also including the flow of the quark two-point function) the system can be understood as the fRG analogues of the complete Bethe-Salpeter equations and quark gap equation.

I think this article will be a further step on the road to a better understanding of QCD.

The manuscript is clear and well written. I just have a few questions for the Authors that appear below (questions that will not imply changes in the presented work).

1-The pion mass is widely studied in section IV-Emergent bound states. The Authors have computed on-shell properties of pions within a Fierz complete computation with momentum-independent coupling except the resonant pseudoscalar channel. I wonder how difficult is it to obtain the pion decay constant? I also wonder the same for the sigma mode: is it too complicated to look at sigma's mass? This can be important for the restoration of chiral symmetry at finite temperature or density.

2-As pointed out by the Authors, for timelike momenta it is expected that further channels, like diquarks, become relevant or even dominating. Once diquarks form a condensate above the threshold is it possible to apply this approach also to diquarks? The same for vector mesons. Although they are not relevant to the study of dynamical chiral symmetry breaking, they are important at finite density.

3-In Fig. 6, for small initial masses M_q<M_{chi} it is detect an unphysical growth of the constituent quark mass M_q on the current quark mass M q,Λ. This is strange to me, even if this occurs in the momentum-independent approximation and even if with dynamical hadronization no unphysical rise of the constituent quark mass occurs in the chiral limit. I do not follow the line of argument as it is currently presented (that the mass and coupling flows are too large due to the lack of momentum-dependence) – could the authors clarify what they mean?

4-How complex is extending this work to Nf=2+1?

5-Do they plan to extend this work to finite temperature and density?

Finally, I appreciated the Authors sincerity when they say that the GMOR relation is violated when M_{q,Λ} is very small, attributing this to the fact that the flows of the quark mass and the four-quark couplings are not solved self-consistently.

By the way, the appendices, where the technical details can be found, are also very good.

In my opinion, the article is suitable for publication and I look forward to read the new articles with developments on Four-quark scatterings in QCD II and III.

Requested changes

I have no substantial changes to ask to be made.
Just a re-read because, as always happens, there are some typos in the text.

  • validity: top
  • significance: high
  • originality: top
  • clarity: top
  • formatting: perfect
  • grammar: excellent

Author:  Chuang Huang  on 2022-12-02  [id 3097]

(in reply to Report 1 on 2022-11-16)

We would like to thank the referee for his positive comments on this manuscript. There are the responses to the questions raised by the referee.

(1) In fact in the second paper of this series we have computed the pion decay constant within an improved truncation, where a momentum-dependent quark mass and three-momentum $s$-, $t$-, $u$-channel dependent four-quark vertices are used. The pion decay constant can be calculated via the definition as follows
\begin{align}
i P_{\mu}f_{\pi}\delta^{ab}&\equiv \langle{0}|J_{5\mu}^{a}(x)| \pi^{b }\rangle =\int \frac{d^{4}q}{(2\pi)^{4}} \mathrm{Tr}\Big{[}\gamma_{\mu}\gamma_{5}T^{a}G_{q}(q+P)h_{\pi}(q,P)\gamma_{5}T^{b}G_{q}(q)\Big{]}
\end{align}
with the Bethe-Salpeter amplitude of pion $h_{\pi}(q,P)$, which can be extracted from the residue of the four-quark coupling at the pole of pion mass. From the equation above, it is clear that in order to calculate $f_{\pi}$ one needs the momentum-dependent quark mass and Bethe-Salpeter amplitude.

Furthermore, the sigma mass is also calculated in Paper II, where the pole mass is calculated by means of analytic continuation based on the results in the Euclidean regime.

(2) It is indeed an interesting question. In principle, this approach can be used to investigate the dynamics of the diquark and vector channels. We also have the plan to investigate the $\rho$ meson within this approach. The off-shell dynamics of diquarks are also accessible with the Fierz-complete four-quark basis in the vacuum or at finite temperature and density, but we are not sure whether the diquark condensate can be calculated with the current setup. We will keep thinking about this question.

(3) As the schematic diagram of meson resonance in Fig.13 shows, when the momentum of pion is in the vicinity of its on-shell momentum, one has the four-quark coupling of the pion channel $\lambda_{\pi}\sim 1/(p^{2}+m_{\pi}^{2})$. In the chiral limit, the pion mass is vanishing, i.e., $m_{\pi} \to 0$, which leaves us with $\lambda_{\pi} \to \infty$ if the momentum dependence is neglected and assumed to be $p=0$. Consequently, the constituent quark mass is enhanced by the overestimated four-quark coupling. It is expected that this effect is especially obvious when the chiral limit is being approached. In Paper II we find the enhancement of the constituent quark mass near the chiral limit is significantly suppressed when more momentum dependence for the four-quark vertex is included.

As for the comparison to the dynamical hadronization, frankly speaking, we have no conclusion yet. We guess it may be attributed to the role played by the potential of mesons in the approach of dynamical hadronization.

(4) In fact, the extension of this approach to $N_{f}=2+1$ has always been in our minds. It is not difficult from the concept side, but obviously it is not readily accessible from the side of computation.

(5) Yes, we have a plan to extend this work to finite temperature and density.

Thanks again for the interesting questions. Moreover, we have corrected some typos in the manuscript in the revised version.

---

## Round 2 · Referee Report · Anonymous (Referee 2) · 2022-12-12

Report

I am completely satisfied with the answers given by the Authors.
In my opinion it can be published.

---

## Editorial Decision

published